# Evolution of fibroblasts in the lung metastatic microenvironment is driven by stage-specific transcriptional plasticity

Ophir Shani[1†], Yael Raz[1,2†], Lea Monteran[1], Ye'ela Scharff[1], Oshrat Levi-Galibov[3], Or Megides[4], Hila Shacham[4], Noam Cohen[1], Dana Silverbush[5], Camilla Avivi[6], Roded Sharan[5], Asaf Madi[1], Ruth Scherz-Shouval[3], Iris Barshack[6], Ilan Tsarfaty[4], Neta Erez[1]*

[1]Department of Pathology, Sackler Faculty of Medicine, Tel Aviv University, Tel Aviv, Israel; [2]Department of Obstetrics and Gynecology, Tel Aviv Sourasky Medical Center, Tel Aviv, Israel; [3]Department of Biomolecular Sciences, The Weizmann Institute of Science, Rehovot, Israel; [4]Department of Clinical Microbiology and Immunology, Sackler Faculty of Medicine, Tel Aviv University, Tel Aviv, Israel; [5]Blavatnik School of Computer Sciences, Faculty of Exact Sciences, Tel Aviv University, Tel Aviv, Israel; [6]Department of Pathology, Sheba Medical Center, Tel Hashomer, affiliated with Sackler Faculty of Medicine, Tel Aviv University, Tel Aviv, Israel

**Abstract** Mortality from breast cancer is almost exclusively a result of tumor metastasis, and lungs are one of the main metastatic sites. Cancer-associated fibroblasts are prominent players in the microenvironment of breast cancer. However, their role in the metastatic niche is largely unknown. In this study, we profiled the transcriptional co-evolution of lung fibroblasts isolated from transgenic mice at defined stage-specific time points of metastases formation. Employing multiple knowledge-based platforms of data analysis provided powerful insights on functional and temporal regulation of the transcriptome of fibroblasts. We demonstrate that fibroblasts in lung metastases are transcriptionally dynamic and plastic, and reveal stage-specific gene signatures that imply functional tasks, including extracellular matrix remodeling, stress response, and shaping the inflammatory microenvironment. Furthermore, we identified *Myc* as a central regulator of fibroblast rewiring and found that stromal upregulation of *Myc* transcriptional networks is associated with disease progression in human breast cancer.

*For correspondence:
Netaerez@tauex.tau.ac.il

†These authors contributed equally to this work

Competing interests: The authors declare that no competing interests exist.

## Introduction

Breast cancer continues to be one of the leading causes of cancer-related death in women, and mortality is almost exclusively a result of tumor metastasis. Advanced metastatic cancers are mostly incurable, and available therapies generally prolong life to a limited extent. It is increasingly appreciated that in addition to tumor cell-intrinsic survival and growth programs, the microenvironment is crucial in supporting metastases formation (*Erez and Coussens, 2011*; *Obenauf and Massagué, 2015*; *Quail and Joyce, 2013*). Nevertheless, while years of research have revealed the central role of the microenvironment in supporting tumor growth and response to therapy at the primary tumor site (*Quail and Joyce, 2013*; *Hanahan and Coussens, 2012*; *Albini et al., 2018*), the role of the metastatic microenvironment and the molecular crosstalk between stromal cells, including fibroblasts and immune cells at the metastatic niche, are poorly characterized.

Preparation of secondary sites to facilitate subsequent tumor cell colonization has been described for multiple cancers (*Peinado et al., 2017*). Secreted factors and extracellular vesicles from tumor and stromal cells were reported to instigate a permissive pre-metastatic niche by influencing the recruitment and activation of immune cells (*Deng et al., 2012*; *Peinado et al., 2011*; *Qian et al., 2011*; *Coffelt et al., 2015*; *Quail et al., 2017*), and by modifying the composition of the extracellular matrix (ECM) (*Erler et al., 2009*; *Malanchi et al., 2012*; *Oskarsson et al., 2011*; *Oskarsson and Massagué, 2012*; *Nielsen et al., 2016*). Each metastatic microenvironment exerts specific functions that support or oppose colonization by disseminated tumor cells (*Peinado et al., 2017*; *Nguyen et al., 2009*). Therefore, understanding distinct organ-specific mechanisms that enable metastatic growth is of crucial importance.

Lungs are one of the most common sites of breast cancer metastasis. Various immune cell populations were shown to be functionally important in facilitating breast cancer pulmonary metastasis (*Coffelt et al., 2015*; *Albrengues et al., 2018*; *DeNardo et al., 2009*; *Fridlender et al., 2015*; *Jablonska et al., 2017*). However, very little is known about the role of fibroblasts during the complex process of metastases formation.

Cancer-associated fibroblasts (CAFs) are a heterogeneous population of fibroblastic cells found in the microenvironment of solid tumors. In some cancer types, including breast carcinomas, CAFs are the most prominent stromal cell type, and their abundance correlates with worse prognosis (*Liu et al., 2016*). We previously demonstrated a novel role for CAFs in mediating tumor-promoting inflammation in mouse and human carcinomas (*Erez et al., 2013*; *Erez et al., 2010*). We further characterized the origin, heterogeneity and function of CAFs in breast cancer (*Sharon et al., 2015*; *Raz et al., 2018*; *Cohen et al., 2017*). Importantly, we found profound changes in the expression of pro-inflammatory genes in fibroblasts isolated from metastases-bearing lungs (*Raz et al., 2018*). However, comprehensive profiling of metastasis-associated fibroblasts in spontaneous metastasis was not previously done. Based on the central role of CAFs in supporting tumor growth at the primary tumor site (*Kalluri, 2016*), we hypothesized that transcriptional reprogramming of lung fibroblasts is an important factor in the formation of a hospitable metastatic niche that supports breast cancer metastasis.

In this study, we set out to characterize the dynamic co-evolution of fibroblasts during pulmonary metastasis. To achieve this goal, we utilized novel transgenic mice that enable visualization, tracking, and unbiased isolation of fibroblasts from spontaneous lung metastases. Here, we demonstrate the profiling and analysis of the dynamic evolution of fibroblast transcriptome at distinct disease stages, including early and late metastatic disease.

## Results

### Fibroblasts are activated and transcriptionally reprogrammed in the lung metastatic niche

We previously demonstrated that fibroblasts at the primary tumor microenvironment are reprogrammed to obtain a pro-inflammatory and tumor-promoting phenotype (*Erez et al., 2010*; *Sharon et al., 2015*; *Cohen et al., 2017*). Moreover, we found that fibroblasts are also modified at the lung metastatic niche (*Raz et al., 2018*). In this study, we set out to characterize the changes in lung fibroblasts that mediate the formation of a hospitable niche in breast cancer metastasis.

We initially investigated metastasis-associated fibroblasts in the lung metastatic microenvironment of MMTV-PyMT transgenic mice with spontaneous lung metastases, compared with normal lungs. We analyzed the changes in the population of fibroblasts using immunostaining with multiple known fibroblast markers including αSMA, FSP-1 (*Gascard and Tlsty, 2016*; *Kalluri and Zeisberg, 2006*), and Podoplanin (PDPN) (*Friedman, 2020*; *Figure 1A–D*). Notably, analysis of αSMA and FSP-1 indicated an upregulation in the expression of these markers in metastases-bearing lungs (*Figure 1B, C*), suggesting that lung metastases are associated with fibroblast activation.

We therefore set out to characterize the changes in fibroblasts at the metastatic niche during the formation of spontaneous lung metastases. To enable visualization, tracking, and isolation of fibroblasts, we established a transgenic mouse model of breast cancer with fibroblast-specific reporter genes: transgenic mice that express the fluorescent reporter YFP under the collagen-1α promoter (*Col1a1*-YFP) were crossed with MMTV-PyMT mice to create PyMT;*Col1a1*-YFP transgenic mice, in

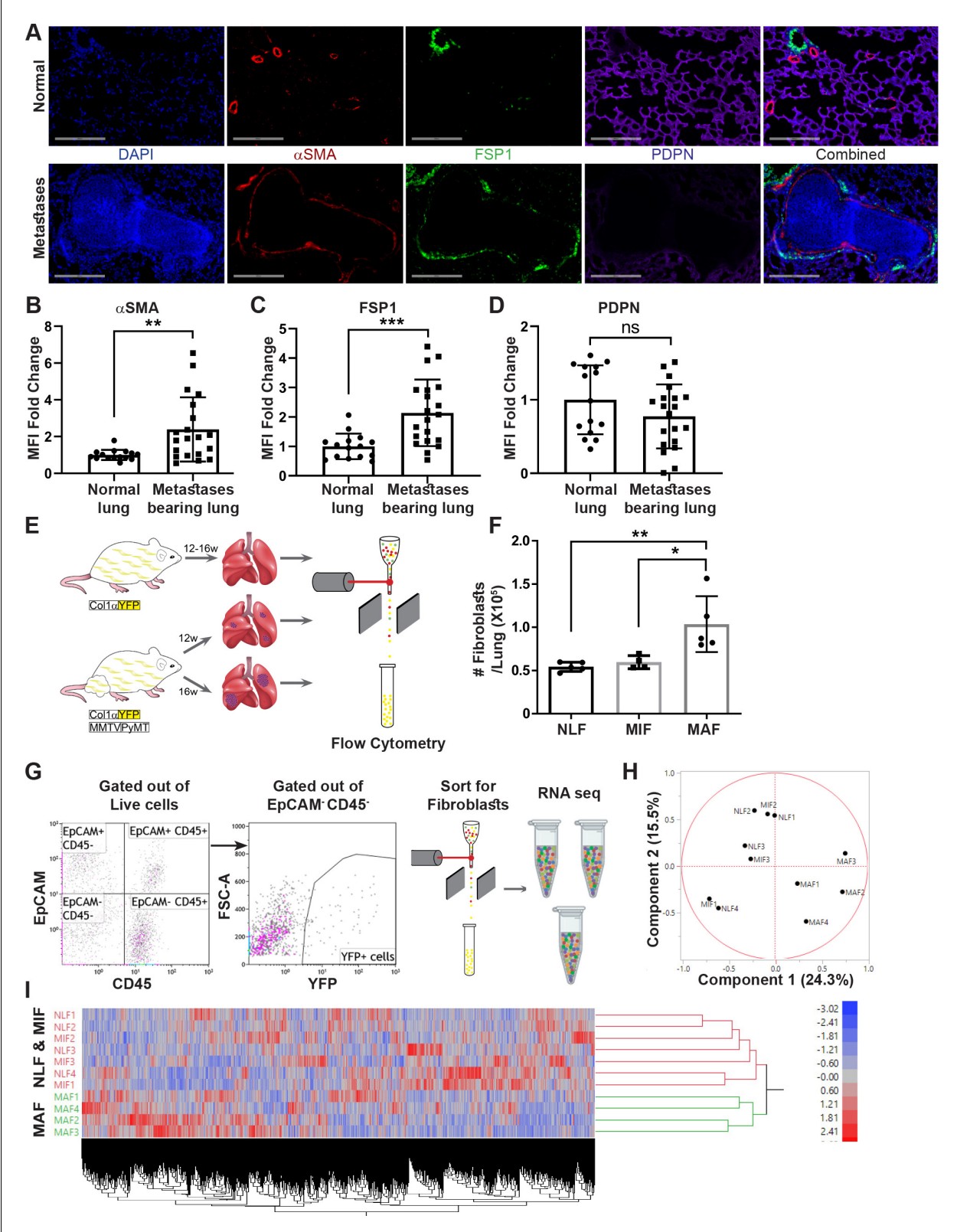

**Figure 1.** Fibroblasts are activated and transcriptionally reprogrammed in the lung metastatic niche. (**A**) Representative immunofluorescent staining of αSMA (red), FSP-1 (green), and Podoplanin (PDPN) (purple) in normal lungs from FVB/n mice (n = 3), and metastases-bearing lungs from MMTV-PyMT mice (n = 4). Scale bar: 200 μM. (**B–D**) Quantification of mean fluorescent intensity (MFI) in 5 fields of view (FOV) per mouse of staining shown in (**A**). (**E**) Workflow illustration of fibroblast isolation (CD45⁻EpCAM⁻YFP⁺) from normal FVB/n;*col1a1*-YFP mice (NLF) and of micro- or macrometastasis-associated

*Figure 1 continued on next page*

Figure 1 continued

fibroblasts from MMTV-PyMT;*Col1a1*-YFP mice (MIF and MAF). (**F**) Quantification of number of fibroblasts per lung, based on flow cytometry analysis. *p<0.05, **p<0.01. Data are represented as mean ± SD, n = 5. (**G**) Flow cytometry gating strategy for isolation of fibroblasts prior to RNA-sequencing. (**H, I**) Principal component analysis (PCA) (**H**) and hierarchical clustering (**I**) of 11,115 protein coding genes identified in RNA-seq.

The online version of this article includes the following figure supplement(s) for figure 1:

**Figure supplement 1.** Volcano plots of differential expression analysis vs. mean expression of micrometastasis-associated fibroblast (MIF) vs. normal lung fibroblast (NLF), macrometastasis-associated fibroblast (MAF) vs. NLF and MAF vs. MIF using DESeq2.

which all fibroblasts are fluorescently labeled (*Raz et al., 2018*). Flow cytometric analysis of normal lungs as compared with lungs of tumor-bearing mice revealed significantly increased numbers of fibroblasts in macrometastatic lungs (*Figure 1E, F*). Thus, fibroblasts are both activated and increase in numbers in the metastatic microenvironment of breast cancer lung metastasis.

To analyze the transcriptional reprograming of activated fibroblasts at the lung metastatic niche, we performed unbiased profiling by RNA-seq of fibroblasts isolated from lungs of PyMT;*Col1a1*-YFP transgenic mice at distinct metastatic stages compared with fibroblasts isolated from normal lungs of *Col1a1*-YFP mice. To explore the temporal changes in functional gene networks, we profiled fibroblasts (EpCAM$^-$CD45$^-$YFP$^+$ cells) isolated from normal lungs, and from lungs with micro- or macrometastases (*Figure 1G*). Micrometastases were defined by the presence of tumor cells in lungs, where no lesions were detectible macroscopically or by CT imaging.

Initial data analysis indicated that fibroblasts isolated from lungs with macrometastases (macrometastasis-associated fibroblasts [MAFs]) were strikingly different from Normal lung fibroblast (NLF) as well as from fibroblasts isolated from lungs with micrometastases (micrometastasis-associated fibroblasts [MIFs]) (*Figure 1H, I*, *Figure 1—figure supplement 1*). Notably, since fibroblasts were isolated from entire lungs, rather than from specific metastatic lesions, the MIF fraction contained a mixture of normal, non-metastasis-associated fibroblasts as well as metastasis-associated fibroblasts. As a result, initial data analysis did not reveal significant differences between NLF and MIF. Thus, metastasis-associated fibroblasts are not only functionally activated but also transcriptionally reprogrammed.

## Transcriptome profiling of metastasis-associated fibroblasts reveals dynamic stage-specific changes in gene expression

In light of these initial results, we next analyzed the genes that are differentially expressed between MAF and NLF. We selected upregulated and downregulated genes based on fold change (FC) of |2|. Expectedly, hierarchical clustering based on these genes revealed that the MAF group clustered separately from NLF and MIF (*Figure 2A*). To better characterize the trajectory of changes in fibroblasts during metastases formation, we next compared the expression of genes that were differentially expressed between MAF and NLF to their expression in the MIF population. Interestingly, we found that the expression pattern in MIF was distinct from both the MAF and the NLF gene expression, including genes that had opposite changes in MAF vs. MIF, suggesting that they are activating a distinct transcriptional program (*Figure 2B*).

We therefore analyzed the differentially expressed genes in the MIF fraction separately. Since the detectible changes in micrometastases were more subtle than the changes detected in the macrometastases group, we selected these genes based on an FC of |1.5| to better differentiate the MIF group from NLF. Indeed, hierarchical clustering based on these differentially expressed genes confirmed that the MIF group clustered separately from both NLF and MAF (*Figure 2C*). Next, we selected a group of genes based on their differential expression between the MAF and MIF groups (FC $\geq$ |2|). The combination of these yielded a total of 897 genes that were differentially expressed in MIF vs. NLF, MAF vs. NLF or MAF vs. MIF. Interestingly, only a small number of these genes were shared across the different stages, suggesting again that each stage is defined by its own specific gene signature (*Figure 2D*). Accordingly, principal component analysis (PCA) and hierarchical clustering applied on the selected gene signature dataset separated each of the metastatic stages (*Figure 2E, F*).

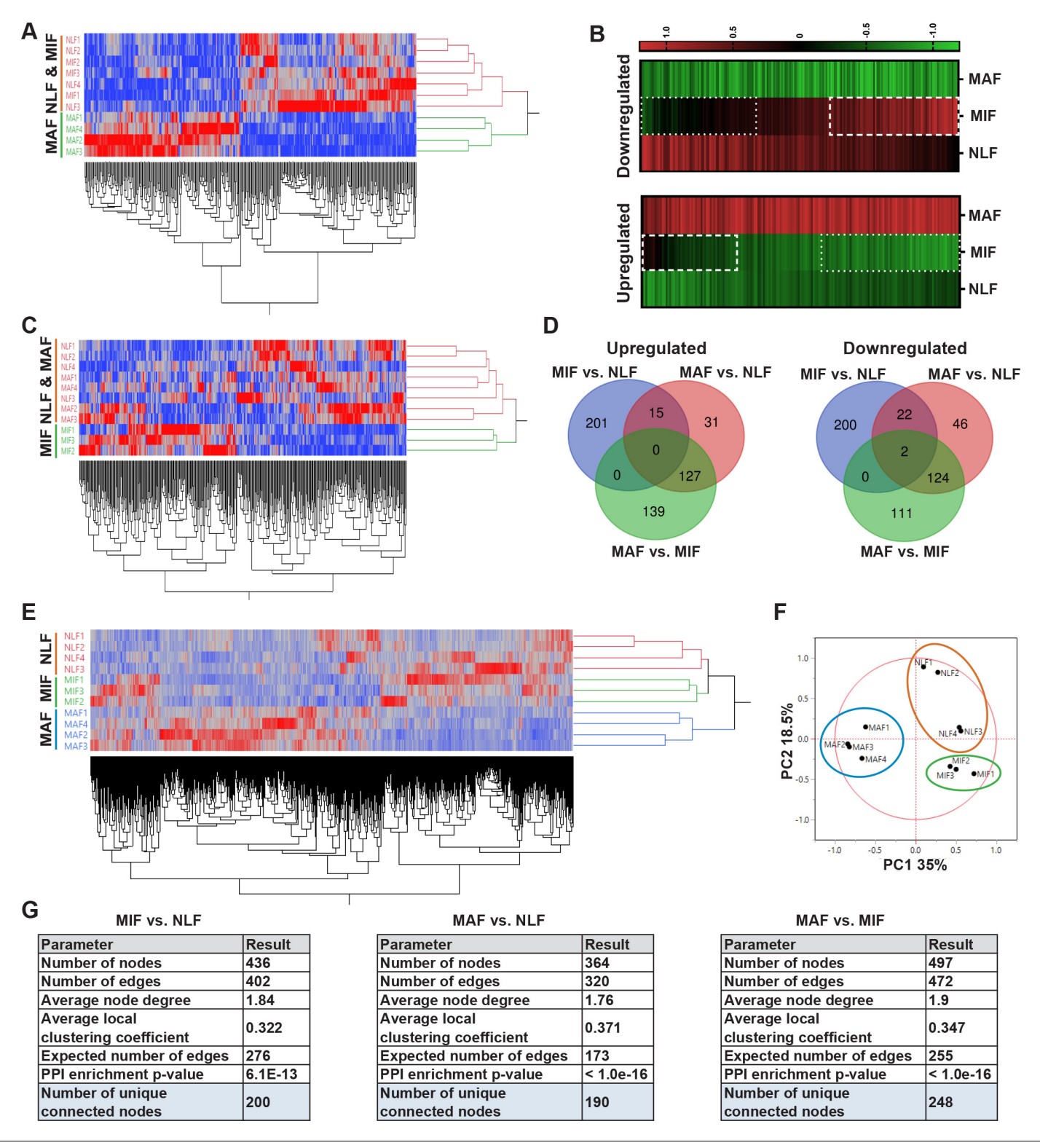

**Figure 2.** Transcriptome profiling of metastasis-associated fibroblasts reveals dynamic stage-specific changes in gene expression. (A) Hierarchical clustering of genes upregulated or downregulated in macrometastasis-associated fibroblast (MAF) vs. normal lung fibroblast (NLF) based on fold change (FC) > |2|. (B) Presentation of the average Z-scored gene expression of genes differentially expression in MAF vs. NLF in all three groups: NLF, micrometastasis-associated fibroblast (MIF) and MAF. Dashed lines demarcate genes upregulated in MIF vs. NLF. Dotted lines demarcate genes downregulated in MIF vs. NLF. (C) Hierarchical clustering of genes upregulated or downregulated in MIF vs. NLF based on FC > |1.5|. (D) Venn diagram

*Figure 2 continued on next page*

*Figure 2 continued*

of upregulated or downregulated genes in the different comparisons. (E, F) Hierarchical clustering (E) and principal component analysis (PCA) (F) of genes upregulated or downregulated in the different comparisons (MIF vs. NLF, MAF vs, NLF, MAF vs. MIF). (G) Protein-protein interaction analysis of the differentially expressed genes per comparison performed in STRING platform. Interconnected genes were selected for subsequent analysis. The online version of this article includes the following figure supplement(s) for figure 2:

**Figure supplement 1.** Protein-protein interactions of differentially expressed genes in each comparison (micrometastasis-associated fibroblast [MIF] vs. normal lung fibroblast [NLF] [1], macrometastasis-associated fibroblast [MAF] vs. NLF [2], MAF vs. MIF [3]), derived from the STRING platform.

Thus, although the transcriptional changes in fibroblasts isolated from micrometastases may have been masked by the presence of normal fibroblasts in this fraction, further analyses suggested that MIF, as well as MAF, activates a unique stage-specific transcriptional program.

Aiming to delineate the stage-specific gene signatures and the molecular mechanisms operative in metastasis-associated fibroblasts, and to identify the most relevant functional pathways, we performed protein-protein interaction (PPI) analysis using the STRING platform (*Szklarczyk et al., 2017*) for each comparison separately. We found that per comparison the differentially expressed genes had significantly more interactions than expected (*Figure 2G*, *Figure 2—figure supplement 1*), suggesting that they are functionally related. We therefore decided to focus our subsequent analyses on the subsets of differentially expressed genes that were found to be inner-connected.

## Fibroblast metastases-promoting features are driven by gene signatures related to stress response, inflammation, and ECM remodeling

We next asked whether the changes in the different metastasis-associated fibroblast subpopulations represent specific metastases-promoting features. To address this question, we performed further analysis of the selected genes in each stage by using the over-representation enrichment analysis of the Consensus Path DB (CPDB) platform (*Kamburov et al., 2011*). Our focus in these analyses was based on three different databases: GO (*Ashburner et al., 2000*; *The Gene Ontology Consortium, 2019*), KEGG (*Kanehisa et al., 2017*; *Kanehisa and Goto, 2000*), and Reactome (*Fabregat et al., 2018*). For our analysis, we selected terms that represent biological processes enriched in at least two databases, with a relative overlap of at least 0.2 and at least two shared entities (*Figure 3A*). Data analysis revealed significant and stage-specific changes in functional pathways including cellular stress response, ECM remodeling, and inflammation (*Figure 3B*, *Supplementary file 1*).

Interestingly, we found that gene expression signatures in fibroblasts isolated from the micrometastatic stage were highly and specifically enriched for functions related to cellular response to stress, including *Hsf1* activation, heat shock response, and response to unfolded protein (*Supplementary file 1*). Upregulated genes in MIF that were related to stress and protein folding included several heat shock proteins: *Hspa8, Hsp90aa1, Hspd1, Hspe1*, and others (*Figure 3C*). Of note, detailed analysis of specific gene expression showed that while the stress response pathway was not significantly enriched in MAF, genes from the stress response pathway were elevated in MAF compared to normal fibroblasts, but not compared to MIF (*Figure 3C*). ECM remodeling terms were enriched in both MIF and MAF (*Figure 3B*), indicating the central importance of ECM modifications in facilitating metastasis. Notably, while ECM remodeling was operative throughout the metastatic process, the specific genes related to ECM remodeling in the different metastatic stages were distinct (*Figure 3D*).

Gene expression signatures in fibroblasts isolated from macrometastases were highly enriched for inflammation-related pathways (*Figure 3B*, *Supplementary file 1*). Indeed, analysis of enriched pathways revealed that genes related to inflammation including many chemokines and cytokines were upregulated specifically in MAF (*Figure 3E*). To validate these findings, we isolated fibroblasts from additional cohorts of mice. We performed qRT-PCR to test the expression of key genes from identified pathways (stress response, ECM remodeling, and inflammation). Analysis of the results confirmed that genes from the identified pathways are specifically upregulated in micro- or macrometastases-associated fibroblasts, in agreement with the RNA-seq results (*Figure 3—figure supplement 1*). Since the MIF population analyzed is highly heterogenous and comprised of tumor-cell-adjacent activated fibroblasts as well as of fibroblasts from tumor-cell-free regions, we also

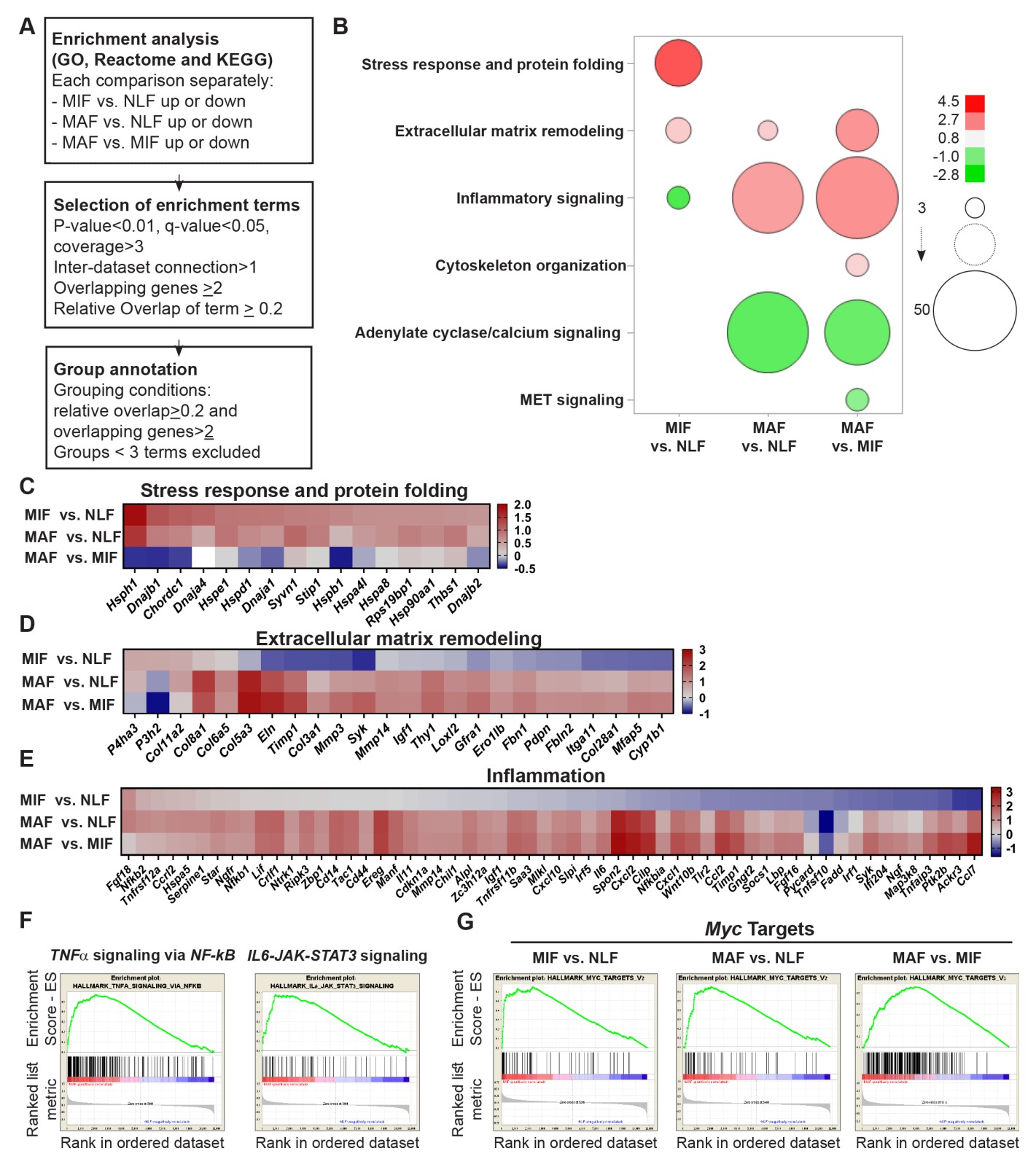

**Figure 3.** Fibroblast metastases-promoting tasks are driven by functional gene signatures related to stress response, inflammation, and extracellular matrix (ECM) remodeling. (**A**) Flow chart of the pathway enrichment over-representation analyses based on GO, Reactome, and KEGG using the Consensus Path DB (CPDB) platform. (**B**) Bubble graph heat map based on the number of specific enrichment terms and their average log-transformed q-value per group. Circle sizes denote number of terms included in a group; color indicates the average log-transformed q-value. Enrichments based

*Figure 3 continued on next page*

*Figure 3 continued*

on downregulated genes are presented as negative values. (**C–E**) Heat maps of gene expression fold-change presenting genes in selected group annotations. Fold change was $\log_2$ transformed for presentation. Only genes found in at least two different terms are presented. (**C**) 'Stress response and protein folding' enriched genes. (**D**) 'Extracellular matrix remodeling' enriched genes. (**E**) 'Inflammatory signaling' and/or 'Cytokine and chemokine activity' enriched genes. (**F**) Gene Set Enrichment Analysis (GSEA) for hallmark datasets upregulated in macrometastasis-associated fibroblast (MAF) vs. normal lung fibroblast (NLF) related to inflammatory signaling, false discovery rate (FDR) < 0.05, normalized enrichment score (NES) > 2. (**G**) GSEA results for '*Myc* targets' hallmark dataset that were upregulated in all comparisons. FDR < 0.05; NES > 2.

The online version of this article includes the following figure supplement(s) for figure 3:

**Figure supplement 1.** qRT-PCR analysis in sorted normal lung fibroblast (NLF), micrometastasis-associated fibroblast (MIF), and macrometastasis-associated fibroblast (MAF).

---

analyzed the spatial expression pattern of two selected genes that were upregulated in the MIF group, THBS1 and HSP90AA1, by immunostaining of lung tissue sections. Staining confirmed that THBS1 and HSP90AA1 are mainly upregulated in MIF. Expectedly, not all YFP$^+$ fibroblasts were THBS1$^+$ or HSP0AA1$^+$, suggesting that MIF are heterogeneous and contain multiple functional sub-populations (*Figure 3—figure supplement 1[2]*).

Taken together, these findings imply that metastasis-associated fibroblasts assume distinct functional roles during the process of lung metastasis.

Encouraged by these findings, we next set out to obtain further insights on functional pathways that were modified in fibroblasts isolated from different metastatic stages. To that end, we performed Gene Set Enrichment Analysis (GSEA) (*Subramanian et al., 2005*). We focused our analysis on the H collection: hallmark gene sets that summarize specific well-defined biological states or processes based on multiple datasets (*Liberzon et al., 2015*). Similar to the results obtained in our previous analyses, we found that functions related to inflammatory responses, including TNFα and IL-6 signaling, were enriched in MAF (*Figure 3F*, *Supplementary file 2*). Interestingly, we found that *Myc* target genes were the most highly and significantly enriched in both metastatic stages (*Figure 3G*, *Supplementary file 2*), suggesting that this transcription factor (TF) may play a central role in the functional molecular co-evolution of metastasis-associated fibroblasts.

Taken together, these findings imply that the transcriptome of lung fibroblasts is rewired during metastatic progression, driving changes in the expression of distinct molecular pathways. Moreover, the transcriptional changes in ECM remodeling and stress response, which represent potential metastases-promoting tasks, are evident at early stages of metastases formation, suggesting that fibroblasts play an important role already at the onset of the metastatic process.

## Multiple gene network analyses identify Myc as a central transcription factor in the rewiring of metastasis-associated fibroblasts

To further characterize the regulatory nodes that govern the transcriptional changes in fibroblasts, we hypothesized that these changes may be driven by TFs related to the pathways that were identified by the pathway and GSEA analyses (*Figure 3*). Analysis of TFs terms within the results identified five candidate TFs that were enriched in at least one analysis and in at least one metastatic stage: *Hif1a, Hsf1, Myc, Nfkb1,* and *Stat3* (*Supplementary file 3*).

We next examined the number of different comparisons in which each TF was enriched. We found that *Hsf1* was only enriched in the micrometastatic stage vs. normal lungs, and *Hif1a* was enriched only in the macrometastatic stage vs. normal lungs. *Nfkb1* and *Stat3* were enriched in the macrometastatic stage compared with both normal and micrometastases. Notably, only *Myc* was enriched in all three comparisons (*Supplementary file 3*).

To rank these TFs, we performed knowledge-based multiple analyses examining their centrality in the selected gene signatures in each comparison (*Supplementary file 4*). We examined the PPIs of these TFs utilizing the STRING platform and counted the number of direct connections of each TF with the metastasis-associated gene signatures. In MAF gene signature, *Stat3* had the largest number of connections, closely followed by *Myc*. In MIF gene signature, *Myc* had the largest number of connections (*Figure 4A*, orange). In addition to STRING, we examined PPIs using ANAT (Advanced Network Analysis Tool) (*Yosef et al., 2011*). In this platform, the inference is based on setting all the candidate TFs as anchors and the selected genes as targets in a network of PPI, and searching for a putative compact subnetwork that connects them. We analyzed the results according to three

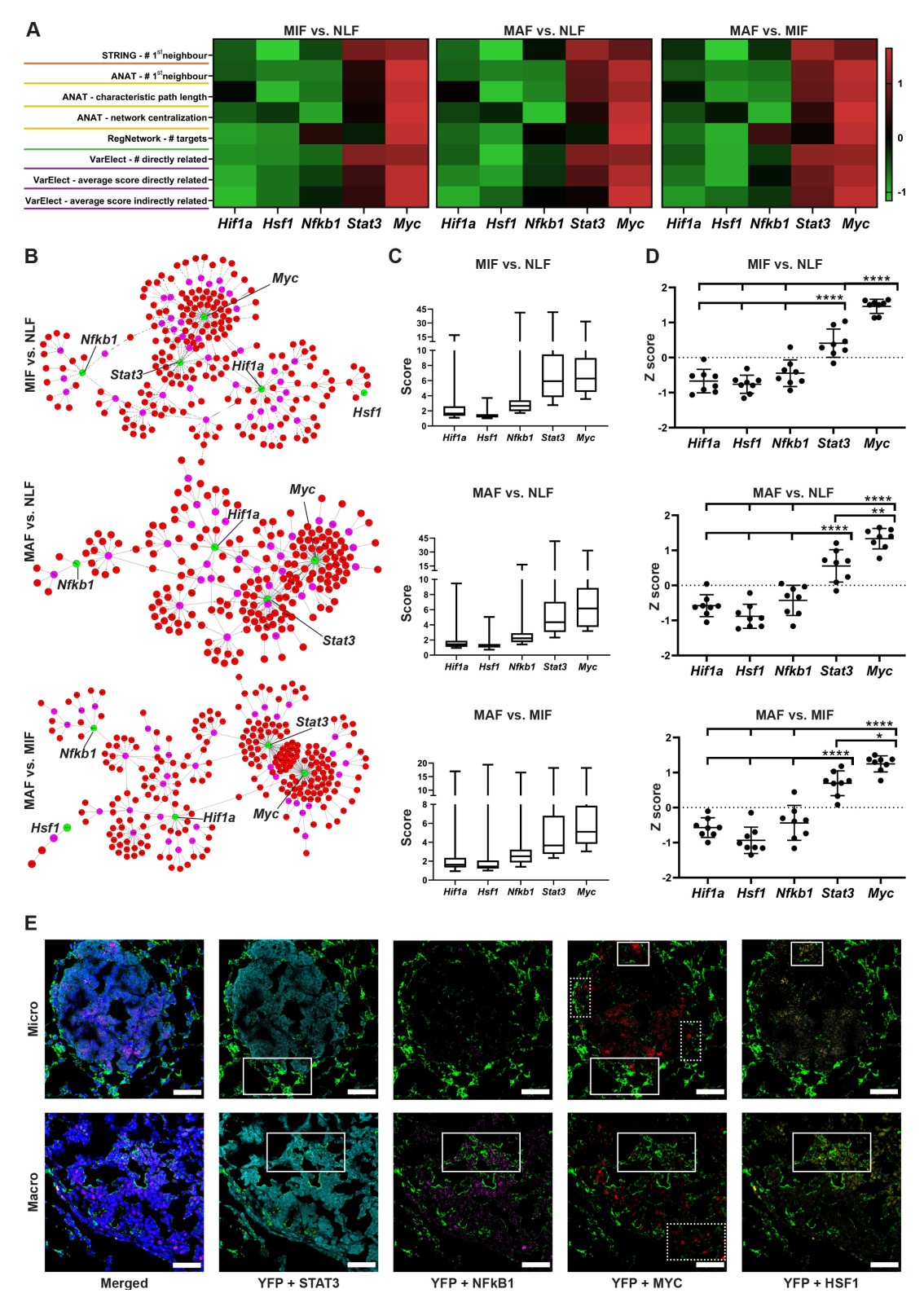

**Figure 4.** Multiple gene network analyses identify Myc as a central transcription factor (TF) in the rewiring of metastasis-associated fibroblasts. (**A**) Heat maps of ranking parameters and analyses performed per each comparison to identify the centrality of five candidate TFs: *Hif1a, Hsf1, Myc, Nfkb1, Stat3*. Orange: STRING protein-protein interaction (PPI) analysis results. Yellow: Advanced Network Analysis Tool (ANAT) pathway analysis results. Green: RegNetwork analysis of connectivity between target genes and TFs. Purple: VarElect analysis results. (**B**) Representative ANAT protein-protein network
*Figure 4 continued on next page*

*Figure 4 continued*

using all TFs as anchors (green) and the stage-specific signature as target genes (red). Only interaction confidence > 0.6 are presented. (C) Box plot of VarElect scores for directly related genes to each TF (presenting top 50 per TF). (D) Z-score graphs of the results described in (A). *p<0.05, **p<0.01, ***p<0.001, ****p<0.0001, one-way ANOVA with Tukey correction for multiple comparisons. Data are presented as mean ± SD. (E) Expression of TFs in micrometastasis-associated fibroblast (MIF) and macrometastasis-associated fibroblast (MAF): representative multiplex immunofluorescent staining (MxIF) staining of YFP (green), STAT3 (cyan), NF-κB (magenta), MYC (red), and HSF1 (yellow) in tissue sections of micro- and macrometastases bearing lungs from PyMT;*Col1a1*-YFP mice (n = 3). Regions with co-staining of several TFs are denoted with solid lines, unique MYC staining regions are denoted in dashed lines. Scale bar: 50 μM.

The online version of this article includes the following figure supplement(s) for figure 4:

**Figure supplement 1.** (1–3) Advanced Network Analysis Tool (ANAT) pathway networks for each transcription factor (TF) (*Hif1a, Hsf1, Myc, Nfkb1, Stat3*) and each comparison (micrometastasis-associated fibroblast [MIF] vs. normal lung fibroblast [NLF] [1], macrometastasis-associated fibroblast [MAF] vs. NLF [2], MAF vs. MIF [3]).

parameters: the number of direct connections of each TF, the characteristic path length to all nodes (including nondirectly related), and network centralization. Analysis of the results revealed that *Myc* had the largest number of direct connections in all comparisons and is overall connected to the fibroblast metastasis-associated gene signatures with the shortest path and with the highest centrality in all comparisons (*Figure 4A*, yellow, *Figure 4B*, *Figure 4—figure supplement 1[1–3]*). These results suggested that *Myc* plays a central role in mediating the transcriptional rewiring of fibroblasts in the lung metastatic niche across the different stages.

We next examined the specific connection of each TF as a regulator in the metastasis-associated gene network. To that end, we utilized the RegNetwork tool (*Liu et al., 2015*), a knowledge-based database of gene regulatory networks. We found that *Myc* had the greatest number of targets in all comparisons, followed by *Stat3* and *Nfkb1* (*Figure 4A*, green). Finally, we analyzed the correlation of the metastasis-associated gene network with each candidate TF using the VarElect tool (*Stelzer et al., 2016*). This tool enables prioritization of genes related to a specific query term by using a direct and indirect relatedness score. We analyzed the scores of the stage-specific signature genes with each candidate TF and the number of directly related genes. The TFs were ranked based on the number and average score for the directly related genes, and the average score of the indirectly related genes. In agreement with previous analyses, *Myc* had the highest number of connections and the highest average score for both directly and indirectly related genes in all comparisons (*Figure 4A*, pink, *Figure 4C*). To consolidate these comprehensive gene network analyses, we performed a comparative analysis on the TF bioinformatics measurements listed in *Figure 4A*. The results indicated that *Myc* achieved significantly higher scores than all other TFs in all three gene signatures (*Figure 4D*).

Since the changes in transcriptome were associated with multiple TFs, we further asked whether the various TFs are co-expressed in the same fibroblasts or in different subpopulations. To address this question, we performed multiplex immunofluorescent staining (MxIF) for YFP, combined with staining for the TFs MYC, STAT3, NFKB1, and HSF1 in lung tissue sections of micro- and macrometastases. Analysis revealed that while some of the fibroblasts co-expressed several TF (*Figure 4E*, solid boxes), other subpopulations expressed only MYC (*Figure 4E*, dashed boxes). Moreover, we found that MYC is expressed in fibroblasts in both micro- and macrometastases. Taken together, these results implicate the putative centrality and unique role of *Myc* in the dynamic transcriptional changes that govern the function of metastasis-associated fibroblasts in lung metastasis.

## Myc is a central regulator in metastasis-associated fibroblasts and contributes to their acquisition of tumor-promoting traits

*Myc* (myelocytomatosis oncogene) is a TF involved in many biological processes, including cell growth and proliferation, cell stemness, and metabolism. *Myc* is deregulated in many human cancers and is known to play an important role in the pathogenesis of cancer, particularly in cancer cells (*Dang, 2012*; *Poole and van Riggelen, 2017*).

To validate the ranking results, we analyzed by qRT-PCR the expression of *Myc* in fibroblasts isolated from normal lungs or from lungs with micro- and macrometastases. Analysis of the results indicated that *Myc* is significantly upregulated in macrometastases-associated fibroblasts (*Figure 5A*). In addition, we assessed the expression of central *Myc* targets that we found to be upregulated in

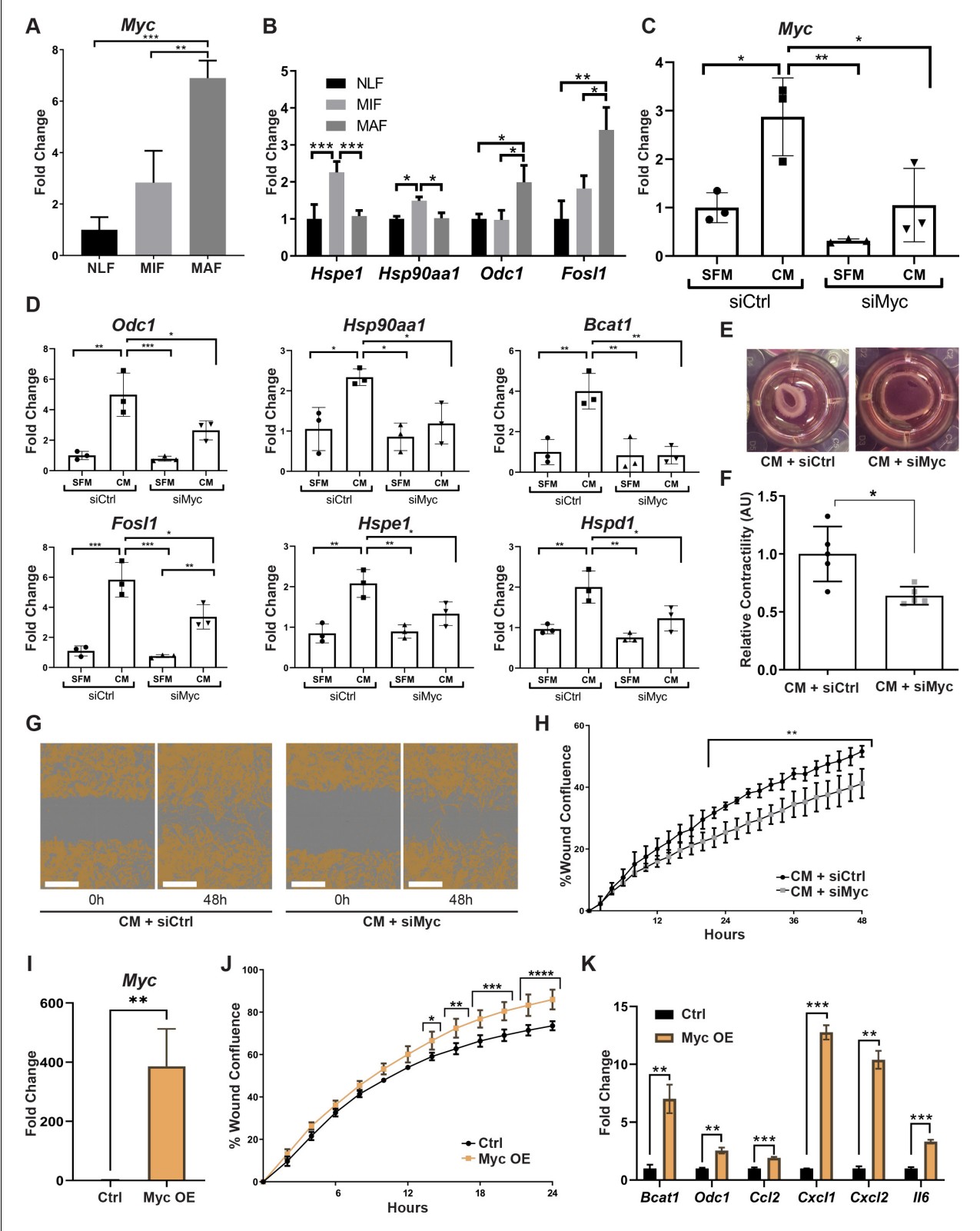

**Figure 5.** Myc is a central regulator in metastasis-associated fibroblasts and contributes to their acquisition of tumor-promoting traits. (**A**) qRT-PCR analysis of *Myc* expression in sorted normal lung fibroblast (NLF), micrometastasis-associated fibroblast (MIF), and macrometastasis-associated fibroblast (MAF). **p<0.01. Data are represented as mean ± SD, n = 3 per group. (**B**) qRT-PCR analysis in sorted NLF, MIF, and MAF. Relative expression of *Myc* target genes found to be differentially expressed in RNA-seq. *p<0.05. Data are presented as mean ± SD, n ≥ 3 per group. (**C**) *Myc*

*Figure 5 continued on next page*

*Figure 5 continued*

targeting by siRNA: *Myc* expression in NLF transfected with siRNA targeting *Myc* or with control siRNA (siMyc or siCtrl). Following transfection, cells were incubated with serum-free medium (SFM) or with Met-1 conditioned media (CM) supplemented with the same siRNA for additional 24 hr. Data are presented as mean ± SD, n = 3. (D) qRT-PCR analysis of *Myc* targets following treatment as in (C). Data are represented as mean ± SD, n = 3. (E, F) Representative images and quantification of collagen contraction assay of fibroblasts transfected with siMyc or siCtrl, incubated with Met-1 CM. *p<0.05. Data are represented as mean ± SD, n = 5. (G, H) Representative images and quantification of scratch closure assay of NLF transfected with siMyc or siCtrl and incubated with Met-1 CM. Scale bar: 400 µm. Two-way ANOVA with multiple comparisons. ***p<0.001. Data are presented as mean ± SD, n = 5. (I) *Myc* overexpression: qRT-PCR analysis of *Myc* expression in NLF transfected with Myc or with a control plasmid (Myc OE or Ctrl). Data are presented as mean ± SD, n = 3. (J) Quantification of scratch closure assay of NLF transfected with Myc or a control plasmid. Two-way ANOVA with multiple comparisons. *p<0.5, **p<0.01, ***p<0.001, ****p<0.0001. Data are presented as mean ± SD, n = 3. (K) qRT-PCR analysis of *Myc* target genes following treatment as in (I). Data are represented as mean ± SD, n = 3.

The online version of this article includes the following figure supplement(s) for figure 5:

**Figure supplement 1.** Representative images of scratch closure assay at 0 hr and 24 hr following scratch.

metastasis-associated fibroblasts, including *Hspe1*, *Hsp90aa1*, *Odc1,* and *Fosl1* (*Belinky et al., 2015*; *Chakravorty et al., 2017*). The results indicated that these *Myc* targets were upregulated in fibroblasts isolated from lungs with metastases (*Figure 5B*). qRT-PCR results of Myc target genes further confirmed that the stress response-related genes *Hsp90aa1* and *Hspe1* were upregulated in MIF, whereas the other *Myc* targets were upregulated in MAF (*Figure 5B*, *Figure 2—figure supplement 1*). To elucidate the functional importance of *Myc* in mediating lung fibroblast reprogramming, we targeted its expression by a specific *Myc* targeting siRNA in primary lung fibroblasts. Abrogation of *Myc* expression by siMyc resulted in significant inhibition of *Myc* expression as compared with control fibroblasts (*Figure 5C*). Importantly, control fibroblasts highly upregulated the expression of *Myc* in response to tumor cell-secreted factors (*Figure 5*C, left bars), while *Myc* inhibition abrogated the upregulation of *Myc* in response to tumor cell-secreted factors in activated fibroblasts (*Figure 5C*, right bars). We next assessed whether inhibition of *Myc* affected the expression of selected *Myc* target genes in activated lung fibroblasts (ALFs). Analysis of the results indicated that targeting the expression of *Myc* significantly inhibited the expression of its target genes in response to tumor cell conditioned media (CM), indicating that the expression of *Myc* in fibroblasts is central to the upregulation of its known targets (*Figure 5D*). Finally, we examined the importance of *Myc* for functional reprogramming of fibroblasts. Fibroblasts at the primary tumor site were previously shown to be reprogrammed by tumor cell-derived paracrine signaling (*Sharon et al., 2015*; *Jin et al., 2017*). We therefore first asked whether fibroblasts at the metastatic microenvironment are similarly activated in response to tumor-secreted factors. Incubation of isolated primary lung fibroblasts with CM from Met-1, a PyMT-derived breast carcinoma cell line (*Borowsky et al., 2005*), or from 4T1 cells, a model of triple-negative breast cancer, indicated that tumor-derived factors activated multiple CAF-associated functions including enhanced motility in wound healing assay (*Figure 5—figure supplement 1[1–4]*) and increased contraction of collagen gel matrices (*Figure 5—figure supplement 1[5–7]*). Thus, NLFs are reprogrammed by signaling from breast cancer cells, resulting in acquisition of tumor-promoting properties. To test whether activation of Myc in lung fibroblasts contributes to their acquisition of CAF characteristics, we performed wound healing assays and collagen contraction assays with tumor-activated lung fibroblasts that were transfected with siMyc or with siCtrl. We found that siMyc fibroblasts were less contractile and exhibited significantly attenuated migration capacity as compared with controls (*Figure 5E–H*, *Figure 5—figure supplement 1[8–9]*). Notably, these changes were not related to any effects of Myc on fibroblast proliferation (*Figure 5—figure supplement 1[10, 11]*).

Since targeting the expression of *Myc* inhibited CAF-like functions of fibroblasts, we next asked whether overexpression of *Myc* would be sufficient to drive fibroblasts into a CAF-like state. NLFs were transduced to overexpress Myc (*Figure 5I*). Interestingly, analysis of CAF-like functions revealed that scratch wound closure was significantly enhanced by overexpression of Myc in a proliferation-independent manner (*Figure 5J*, *Figure 5—figure supplement 1[12]*). Notably, Myc overexpression induced upregulation of its target genes BCAT1 and ODC1, which were also upregulated in MAF. Moreover, multiple pro-inflammatory genes were upregulated by Myc overexpression (*Figure 5K*). While these genes are not direct targets of Myc, they are known NFKB1 target genes. Myc itself is a target of NFKB1 (*Grumont et al., 2004*; *La Rosa et al., 1994*), and the two TFs share

target genes (*Han et al., 2018*). Thus, overexpression of Myc was sufficient to activate CAF-like functions including wound closure and expression of its target genes, as well as pro-inflammatory signaling in fibroblasts.

Taken together, our findings imply that *Myc* has a central role in enhancing fibroblast activation and in mediating their acquisition of metastasis-promoting functions.

## High expression of MYC and its downstream target genes is associated with tumor aggressiveness in human breast cancer

Encouraged by these findings, we next asked whether stromal activation of *MYC* and its downstream targets is operative in human breast cancer. There are currently no available transcriptomic datasets of lung metastases, and we therefore analyzed patient data from breast tumors utilizing a publicly available dataset (*Ma et al., 2009*). Since we showed that *MYC* is a central regulator of fibroblast rewiring during metastatic progression in mice, we asked whether *MYC* is similarly upregulated in the stromal compartment of human breast cancer. Importantly, analysis revealed that *MYC* is upregulated in breast cancer stroma in correlation with disease progression, as reflected by pathological grade: expression of *MYC* was significantly elevated in the stroma of grade 3 tumors compared with stroma isolated from more differentiated tumors (*Figure 6A*). Interestingly, *NFKB1* and *STAT3* did not exhibit this grade-dependent trend of expression (*Figure 6B, C*). To further assess whether the upregulation of stromal *MYC* and its target genes is operative in the stromal compartment of human breast tumors, we compared the expression of *MYC* with the expression of its target genes in human breast cancer patients. Target genes were selected based on their upregulation in metastasis-associated fibroblasts. We found that stromal expression of *MYC* was positively correlated with stromal expression of multiple target genes (*Figure 6D*). Notably, among the *MYC* downstream target genes that were positively correlated with its expression in human patients were several of the genes that were also validated in murine lung fibroblasts: *HSP90AA1*, *HSPD1*, *ODC1,* and *HSPE1* (*Figure 6D*, *Figure 6—figure supplement 1*), suggesting that stromal *MYC*-driven gene signatures play a functional role in human breast cancer. Finally, to validate our findings in human metastasis, we analyzed the expression of MYC in a cohort of breast cancer patients with lung metastasis. We found that MYC was expressed in a subset of lung metastasis-associated stromal cells (*Figure 6E*), suggesting that stromal upregulation of MYC plays a functional role in human lung metastasis.

These results suggest that the activation of MYC transcriptional networks in the stroma of breast tumors plays a role in tumor aggressiveness in human breast cancer.

## Discussion

In this study, we set out to elucidate the dynamic changes in the stromal compartment that facilitate the formation of a hospitable metastatic niche during breast cancer metastasis to lungs. We utilized a unique model of transgenic mice that enabled unbiased isolation of fibroblasts from spontaneous lung metastasis and performed comprehensive analysis of the transcriptome of fibroblasts isolated from normal lungs, and lungs with micro- or macrometastases. By employing multiple platforms of data analysis, we integrated ontology analyses with data on protein interactions and functional pathways from knowledge-based databases to identify the relevant and stage-specific gene signatures that imply functional tasks of metastasis-associated fibroblasts.

Importantly, we performed the analysis on fibroblasts isolated directly from fresh tissues, with no additional culture step that may affect gene expression. Our findings indicated that ECM remodeling programs were instigated early in micrometastases and persisted to be functional throughout metastatic progression, while other signaling pathways were activated in a stage-specific manner. Activation of the cellular stress response was associated with micrometastases, and inflammatory signaling was instigated in fibroblasts isolated from advanced metastases, suggesting that fibroblasts are transcriptionally dynamic and plastic, and that they adapt their function to the evolving microenvironment (*Figure 7*).

Initial analysis of the RNA-seq data revealed distinct gene signatures associated with advanced metastatic disease. By performing step-by-step analysis, a unique gene signature was revealed for early metastatic disease as well. Moreover, utilizing a combination of analyses platforms, we unraveled multiple pathways operative in fibroblasts in different metastatic stages, relying not only on altered gene expression but also on functional role and interaction of genes.

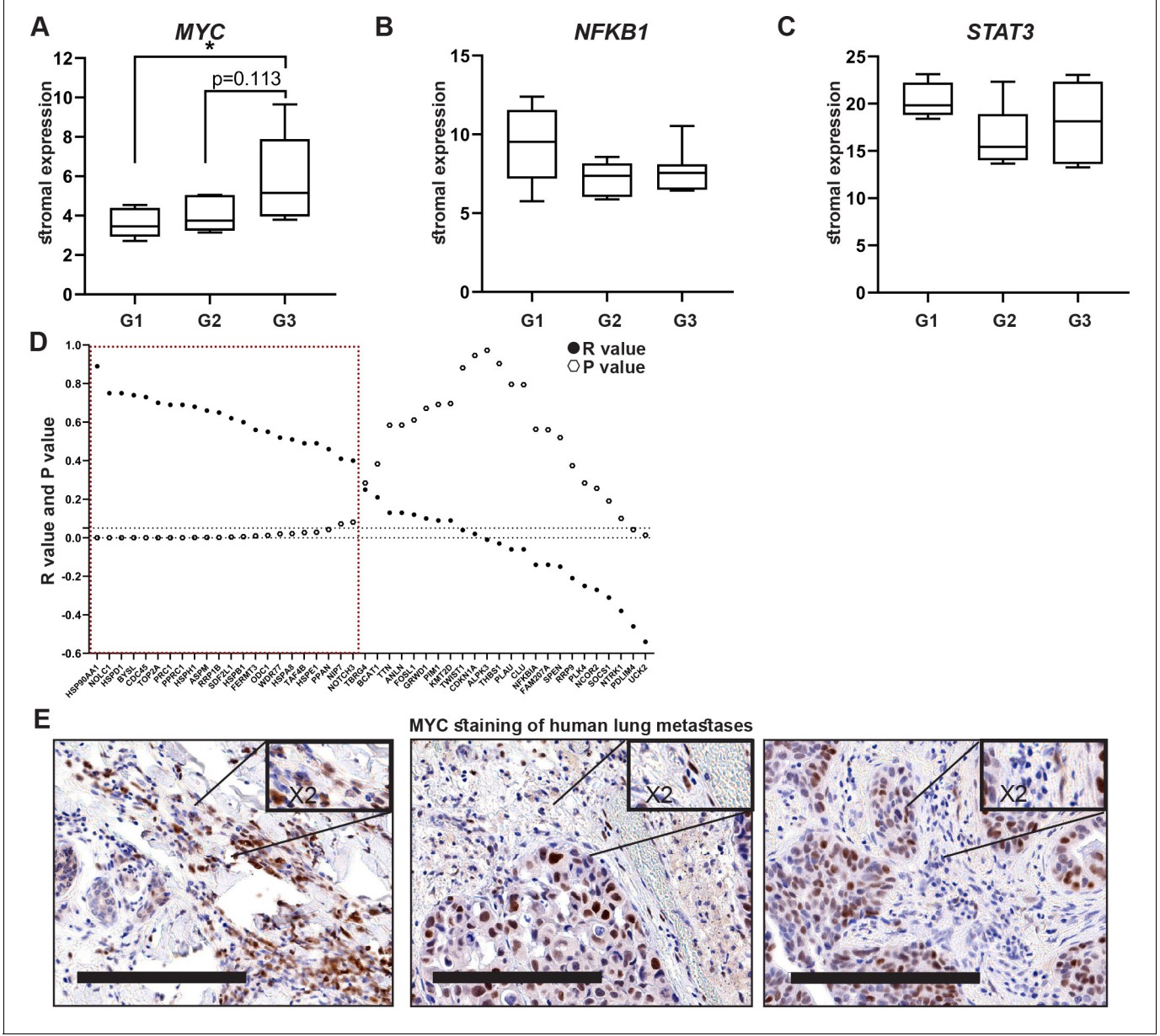

**Figure 6.** High expression of MYC and its downstream target genes is associated with tumor aggressiveness in human breast cancer. (A–C) Box plots of *MYC* (A), *NFKB1* (B), and *STAT3* (C) expression in tumor-associated stroma from the GSE14548 dataset by disease grade (grade 1: G1; grade 2: G2; grade 3: G3). Data are presented as median and upper and lower quartiles ± SD. One-way ANOVA with Tukey correction for multiple comparisons, *p<0.05. (D) Correlations between the expression of *MYC* and selected downstream targets in tumor-associated stroma based on GSE14548. Positive correlations are marked in dotted red square. *p-value<0.05. (E) Representative immunohistochemistry staining of MYC in lung metastases of breast cancer patients (n = 9). Scale bars: 200 μm.

The online version of this article includes the following figure supplement(s) for figure 6:

**Figure supplement 1.** Correlation graphs between *MYC* expression and the expression of specific target genes.

Interestingly, this multilayered analysis indicated that fibroblasts isolated from micrometastases instigated the expression of genes related to cellular response to stress, including the transcriptional regulator *Hsf1*. *Hsf1* was previously shown to be upregulated in CAFs in breast and lung cancers and to drive a stromal tumor-promoting transcriptional program that correlated with worse prognosis (*Scherz-Shouval et al., 2014*). Moreover, *Hsf1* was recently implicated in mediating the transition from chronic inflammation to colon cancer by mediating ECM remodeling (*Levi-Galibov et al.,*

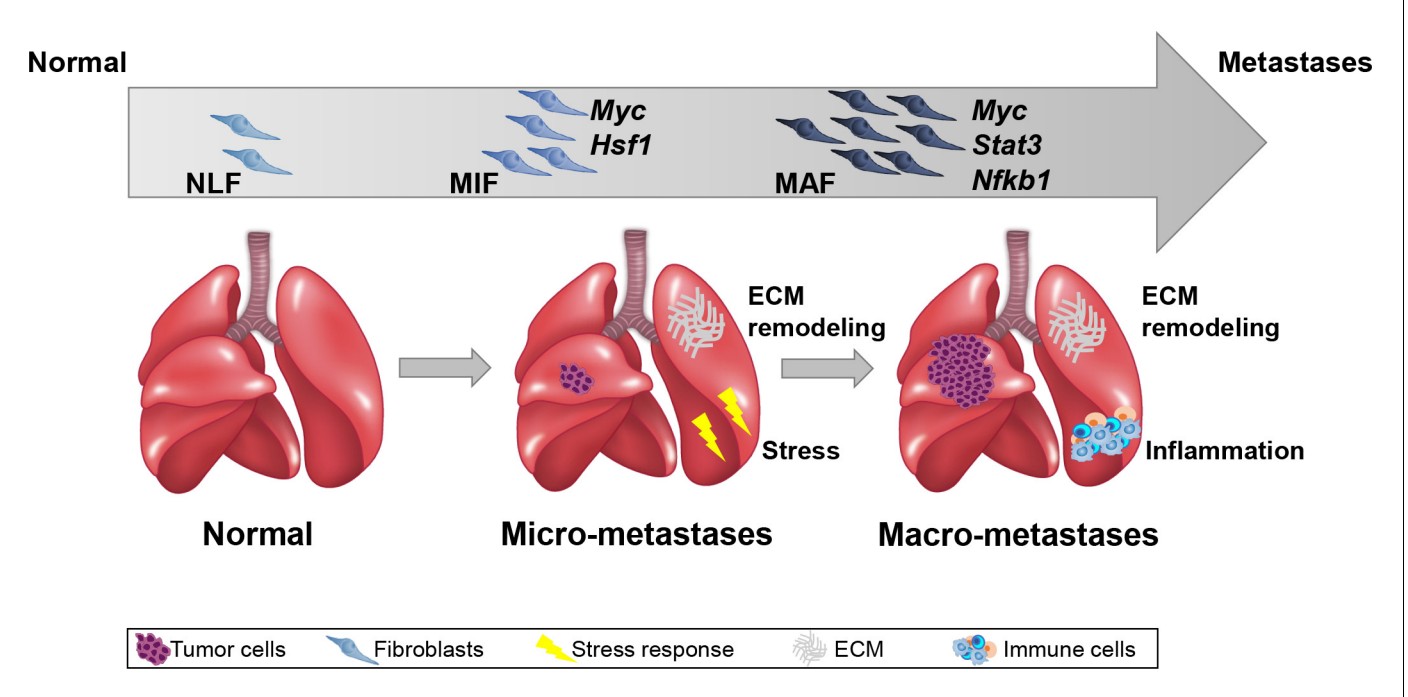

**Figure 7.** Summary scheme. The co-evolution of lung fibroblasts at the metastatic microenvironment is driven by stage-specific transcriptional plasticity that activates growth-promoting tasks including stress response, extracellular matrix (ECM) remodeling, and instigation of inflammatory signaling.

*2020*). Our findings expand these observations to the metastatic microenvironment and show that activation of *Hsf1* transcriptional regulation in fibroblasts occurs during the early stages of metastasis and thus may play a role in instigating tumor-promoting functions in metastasis-associated fibroblasts.

In addition to stress response, our findings indicated that ECM remodeling is a central task of metastasis-associated fibroblasts throughout the metastatic cascade. Indeed, ECM components and remodeling were demonstrated to facilitate breast cancer metastasis to lungs, and pancreatic cancer metastasis to liver (*Malanchi et al., 2012*; *Oskarsson et al., 2011*; *Nielsen et al., 2016*; *Cox et al., 2013*; *Yuzhalin et al., 2018*; *Alexander and Cukierman, 2020*). We show that transcriptional rewiring of fibroblasts to mediate collagen synthesis and ECM organization is a central function of metastasis-associated fibroblasts, which is instigated early during the metastatic process and persists during advanced metastatic disease.

Notably, analyzing the central pathways in fibroblasts that were isolated from advanced metastases indicated that metastasis-associated fibroblasts upregulated pro-inflammatory pathways including multiple cytokines and chemokines. CAFs are known to play a central role in mediating tumor-promoting inflammation at the primary tumor site (*Servais and Erez, 2013*). Importantly, activation of inflammation was also implicated in shaping of the metastatic microenvironment (*Coffelt et al., 2015*; *Quail et al., 2017*; *Albrengues et al., 2018*), but the role of fibroblasts in mediating inflammation at the metastatic site is only recently emerging: recent studies implicated CAF-derived cytokines including IL-1β, IL-33, and CXCL9/10 in promoting breast cancer lung metastasis (*Pein et al., 2020*; *Ershaid et al., 2019*; *Shani et al., 2020*). However, a comprehensive profiling of metastases-associated fibroblasts isolated from spontaneous metastasis in immune competent mice was not previously done.

We further characterized the molecular mechanisms operative in metastasis-associated fibroblasts by identifying the central TFs that drive the metastasis-associated gene programs upregulated in lung fibroblasts. Our analyses revealed several central regulators that are operative in metastasis-associated fibroblasts, including the well-known modulators of CAF activity *Nfkb1* (*Erez et al., 2013*; 24) and *Stat3* (*Chakraborty et al., 2017*; *Li et al., 2018*).

Surprisingly, the most prominent regulator in the metastasis-associated fibroblasts network was the TF *Myc*. While the importance of *Myc* in promoting cell transformation and tumorigenesis is well established (*Poole and van Riggelen, 2017*), its role in the tumor stroma is largely uncharacterized. *Myc* expression in tumor cells was recently shown to be regulated by microenvironmental signals (*Bhattacharyya et al., 2020*) and to drive an inflammatory and immunosuppressive microenvironment (*Kortlever et al., 2017*). Moreover, the expression of *Myc* in the stromal compartment was suggested to mediate metabolic and transcriptional reprogramming of fibroblasts (*Yan et al., 2018*; *Minciacchi et al., 2017*). Our study identifies *Myc* as a central regulator in the transcriptional plasticity of metastasis-associated fibroblasts. Indeed, inhibition of *Myc* attenuated tumor-promoting functions of fibroblasts and overexpression of *Myc* was sufficient to induce these functions, confirming that *Myc* functionally contributes to fibroblast acquisition of tumor-promoting traits. Importantly, validation of these findings in human breast cancer patients revealed that stromal expression of *Myc* and its downstream genes is correlated with disease progression in breast cancer patients. Stromal gene expression was previously found to be associated with bad prognosis in colon cancer (*Calon et al., 2015*). Our findings implicate activation of *Myc* and stromal gene expression in breast cancer patient survival. Taken together, these findings indicate that in addition to its known role in driving carcinogenesis in tumor cells, *Myc* functions in stromal rewiring in the tumor microenvironment in both primary tumors and metastases of breast cancer.

In summary, we show that integration of multiple analytical platforms of gene expression, connectivity, and function provided a powerful insight on functional and temporal regulation of the dynamic transcriptome of fibroblasts in lung metastasis. We uncovered central molecular pathways that drive the activation of growth-promoting tasks in fibroblasts via known regulators of CAF tumor-promoting activities including *Myc*, a novel regulator of fibroblast metastases-promoting properties. Our findings elucidate for the first time the dynamic transcriptional co-evolution of fibroblasts during the multistage process of breast cancer metastasis.

# Materials and methods

## Key resources table

| Reagent type (species) or resource | Designation | Source or reference | Identifiers | Additional information |
|---|---|---|---|---|
| Cell line (*Mus musculus*) | Met-1 | Collaborator's lab | | |
| Cell line (*M. musculus*) | 4T1 | Collaborator's lab | | |
| Transfected construct (*M. musculus*) | siRNA to Myc (Accell SMARTpool) | Dharmacon/ Thermo Fisher Scientific | E-040813 | |
| Transfected construct (*M. musculus*) | siRNA to Myc 1 | Dharmacon/ Thermo Fisher Scientific | A-040813-17 | CCUCAAACUUAAAUAGUAU |
| Transfected construct (*M. musculus*) | siRNA to Myc 2 | Dharmacon/ Thermo Fisher Scientific | A-040813-20 | CUCUGGUGCAUAAACUGAC |
| Transfected construct (*M. musculus*) | siRNA to Myc 3 | Dharmacon/ Thermo Fisher Scientific | A-040813-18 | GCUUCAGCCAUAAUUUUAA |
| Transfected construct (*M. musculus*) | Mouse Myc cDNA pCMV-SPORT6 | Tamar Laboratories | #MMM1013-202763479 | |
| Antibody | Monoclonal rat anti -mouse EpCAM-APC | eBioscience/ Thermo Fisher Scientific | 17-5791 | 1:100 |
| Antibody | Monoclonal rat anti -mouse CD45-PercpCy5.5 | eBioscience/ Thermo Fisher Scientific | 45-0451 | 1:200 |
| Antibody | Monoclonal rat anti -mouse CD31 PeCy7 | eBioscience/ Thermo Fisher Scientific | 25-0311 | 1:50 |
| Antibody | Monoclonal rat anti -mouse Ki67-PE | BioLegend | 652403 | 1:100 |
| Antibody | Monoclonal rabbit anti-mouse Nfkb1 | Cell Signaling | CST-8242S | 1:200 |

*Continued on next page*

*Continued*

| Reagent type (species) or resource | Designation | Source or reference | Identifiers | Additional information |
|---|---|---|---|---|
| Antibody | Monoclonal rabbit anti-mouse HSP90aa1 | Cell Signaling | CST-4877S | 1:200 |
| Antibody | Monoclonal rabbit anti-mouse Stat3 | Cell Signaling | CST 12640S | 1:200 |
| Antibody | Polyclonal chicken anti-GFP/YFP | Abcam | AB-ab13970 | 1:400 |
| Antibody | Polyclonal rabbit anti-GFP/YFP | Abcam | AB-ab6556 | 1:100 |
| Antibody | Monoclonal rabbit anti-mouse Myc | Abcam | AB-ab32072 | 1:200 |
| Antibody | Monoclonal rabbit anti-mouse THBS1 | Abcam | AB-ab263905 | 1:50 |
| Antibody | Polyclonal rabbit anti-mouse Hsf1 | Cell Signaling | 4356S | 1:800 |
| Antibody | Monoclonal mouse anti-mouse aSMA | Sigma-Aldrich | A2547 | 1:1000 |
| Antibody | Polyclonal goat anti-mouse PDPN | R&D Systems | AF3244 | 1:200 |
| Antibody | Polyclonal rabbit anti-mouse FSP-1 (S100A4) | Abcam | Ab41532 | 1:600 |
| Antibody | Polyclonal goat anti-rabbit | Jackson | 111-035-144 | 1:400 |
| Commercial assay or kit | Opal 520 Reagent Pack | Akoya Biosciences | FP1487001 KT | 1:400 |
| Commercial assay or kit | Opal 570 Reagent Pack | Akoya Biosciences | FP1488001 KT | 1:400 |
| Commercial assay or kit | Opal 620 Reagent Pack | Akoya Biosciences | FP1495001 KT | 1:400 |
| Commercial assay or kit | Opal 650 Reagent Pack | Akoya Biosciences | FP1496001 KT | 1:400 |
| Commercial assay or kit | Opal 690 Reagent Pack | Akoya Biosciences | FP1497001 KT | 1:400 |
| Commercial assay or kit | Intracellular staining Kit | BD Biosciences | 554714 | |
| Sequence-based reagent | Bcat1_F | HyLabs | PCR primers | CCCATCGTACCTCTTTCACCC |
| Sequence-based reagent | Bcat1_R | HyLabs | PCR primers | GGGAGCGTGGGAATACGTG |
| Sequence-based reagent | Ccl7_F | HyLabs | PCR primers | CCTGGGAAGCTGTTATCTTCAA |
| Sequence-based reagent | Ccl7_R | HyLabs | PCR primers | GGTTTCTGTTCAGGCACATTTC |
| Sequence-based reagent | Chi3l1_F | HyLabs | PCR primers | GGCAGAGAGAAACTCCTGCTCA |
| Sequence-based reagent | Chi3l1_R | HyLabs | PCR primers | TGAGATTGATAAAATCCAGGTGTTG |
| Sequence-based reagent | Myc_F | HyLabs | PCR primers | CGGACACACAACGTCTTGGAA |
| Sequence-based reagent | Myc_R | HyLabs | PCR primers | AGGATGTAGGCGGTGGCTTTT |
| Sequence-based reagent | Col5a3_F | HyLabs | PCR primers | AGGGACCAACTGGGAAGAGT |

*Continued on next page*

*Continued*

| Reagent type (species) or resource | Designation | Source or reference | Identifiers | Additional information |
|---|---|---|---|---|
| Sequence-based reagent | Col5a3_R | HyLabs | PCR primers | AAAGTCAGAGGCAGCCACAT |
| Sequence-based reagent | Col8a1_F | HyLabs | PCR primers | GCCAGCCAAGCCTAAATGTG |
| Sequence-based reagent | Col8a1_R | HyLabs | PCR primers | GTAGGCACCGGCCTGAATGA |
| Sequence-based reagent | Cxcl10_F | HyLabs | PCR primers | CACCATGAACCCAAGTGCTG |
| Sequence-based reagent | Cxcl10_R | HyLabs | PCR primers | TTGCGAGAGGGATCCCTTG |
| Sequence-based reagent | Fosl1_F | HyLabs | PCR primers | CCAGGGCATGTACCGAGACTA |
| Sequence-based reagent | Fosl1_R | HyLabs | PCR primers | TGGCACAAGGTGGAACTTCTG |
| Sequence-based reagent | Gapdh_F | HyLabs | PCR primers | TGTGTCCGTCGTGGATCTGA |
| Sequence-based reagent | Gapdh_R | HyLabs | PCR primers | TTGCTGTTGAAGTCGCAGGAG |
| Sequence-based reagent | Hsp90aa1_F | HyLabs | PCR primers | GCGTGTTCATTCAGCCACGAT |
| Sequence-based reagent | Hsp90aa1_R | HyLabs | PCR primers | ACTGGGCAATTTCTGCCTGA |
| Sequence-based reagent | Hspd1_F | HyLabs | PCR primers | CACAGTCCTTCGCCAGATGAG |
| Sequence-based reagent | Hspd1_R | HyLabs | PCR primers | CTACACCTTGAAGCATTAAGGCT |
| Sequence-based reagent | Hspe1_F | HyLabs | PCR primers | AGTTTCTTCCGCTCTTTGACAG |
| Sequence-based reagent | Hspe1_R | HyLabs | PCR primers | TGCCACCTTTGGTTACAGTTTC |
| Sequence-based reagent | Hsph1_F | HyLabs | PCR primers | CAACAGAAAGCTCGGATGTGGATAA |
| Sequence-based reagent | Hsph1_R | HyLabs | PCR primers | CTTCTGAGGTAAGTTCAGGTGAAG |
| Sequence-based reagent | Il6_F | HyLabs | PCR primers | ATACCACTCCCAACAGACCTGTCT |
| Sequence-based reagent | Il6_R | HyLabs | PCR primers | CAGAATTGCCATTGCACAACTC |
| Sequence-based reagent | Gusb_F | HyLabs | PCR primers | GCAGCCGCTACGGGAGTC |
| Sequence-based reagent | Gusb_R | HyLabs | PCR primers | TTCATACCACACCCAGCCAAT |
| Sequence-based reagent | Odc1_F | HyLabs | PCR primers | GACGAGTTTGACTGCCACATC |
| Sequence-based reagent | Odc1_R | HyLabs | PCR primers | CGCAACATAGAACGCATCCTT |
| Sequence-based reagent | Timp1_F | HyLabs | PCR primers | GTGCACAGTGTTTCCCTGTTTA |
| Sequence-based reagent | Timp1_R | HyLabs | PCR primers | GACCTGATCCGTCCACAAAC |
| Other | DAPI stain | Molecular Probes | D3571 | 1:1000 |
| Other | DAPI stain | BioLegend | 422801 | 1:1000 |
| Software, algorithm | JMP14 and up | JMP | | |

## Mice

All experiments were performed using 6- to 8-week-old female mice, unless otherwise stated. All experiments involving animals were approved by the Tel Aviv University Institutional Animal Care and Use Committee. FVB/n *Col1a1*-YFP mice were a kind gift from Dr. Gustavo Leone. FVB/N-Tg (MMTV-PyMT) 634Mul/J were backcrossed with FVB/n;*Col1a1*-YFP mice to create PyMT;*Col1a1*-YFP double-transgenic mice as described previously (*Raz et al., 2018*). Non-transgenic Balb/c mice were purchased from Harlan, Israel. All animals were maintained within the Tel Aviv University Specific Pathogen Free (SPF) facility.

## Cell cultures
### Cancer cell lines

Met-1 mouse mammary gland carcinoma cells were a gift from Prof. Jeffrey Pollard. Met-1 cells were plated on 100 mm plastic dishes and cultured with DMEM medium supplemented with 10% FCS, 1% penicillin-streptomycin, and 1% sodium-pyruvate (Biological Industries). 4T1 mouse mammary cell lines were obtained from the laboratory of Dr. Zvi Granot. 4T1 cells were plated on 100 mm plastic dishes and cultured with RPMI medium supplemented with 10% FCS, 1% penicillin-streptomycin, and 1% sodium-pyruvate (Biological Industries). Cell lines were not authenticated in our laboratory. All cell lines were routinely tested for mycoplasma using the EZ-PCR-Mycoplasma test kit (Biological Industries; 20-700-20).

### Primary lung fibroblasts cultures

Lungs were isolated from 6- to 8-week-old FVB/n female mice or Balb/C female mice. Single-cell suspensions were prepared as previously described (*Sharon et al., 2013*). Single-cell suspensions were seeded on 6-well plates pre-coated with Rat tail collagen (Corning; 354236). Cells were grown in DMEM media supplemented with 10% FCS and maintained at 37°C with 5% $CO_2$.

## Conditioned media
### Tumor cell CM (Met-1 CM or 4T1 CM)

Cells were cultured as described above. When cells reached 80% confluency, plates were washed twice with PBS and fresh serum-free medium (SFM) was applied. After 48 hr, medium was collected, filtered through 0.45 μm filters under aseptic conditions, flash-frozen in liquid nitrogen, and stored at −80°C. SFM supplemented as above was used as control.

NLF or ALFs CM

NLFs were plated as described above. CM was prepared by incubating NLF with either SFM (for NLF CM) or tumor cell CM (for ALF CM) for 24 hr. After 24 hr, plates were washed twice with PBS and cells were incubated for additional 24 hr with fresh SFM. After 48 hr, medium was collected, filtered through 0.45 μm filters under aseptic conditions, flash-frozen in liquid nitrogen, and stored at −80°C.

## Scratch assay

NLF were plated in a 96-well IncuCyte imageLock plate (Essen BioScience). SFM was applied for 16 hr. Wells were then washed three times with PBS and a scratch was made using the IncuCyte WoundMaker (Essen Bioscience). Wells were washed three times with PBS and cancer cell CM or SFM were applied. The plate was placed in the IncuCyte system (Essen BioScience) for 48 hr. Images were analyzed using the IncuCyte software. Inhibition of proliferation was performed by adding 20 μg/ml mitomycin C (Sigma-Aldrich; M4287) to all wells during the scratch closure time.

## Collagen contraction

NLFs were plated as mentioned above and incubated with SFM for 16 hr. Following, cells were detached from dishes with trypsin and counted. A total of $1.5 \times 10^5$ fibroblasts were suspended in a medium and collagen mixture (cancer cell CM or SFM mixed with High Concentration Rat Tail Collagen, type 1, BD Biosciences) and allowed to set at 37°C for 45 min. Tumor cell CM or SFM were

applied, gels were released, and incubated for 24 hr. Gels were photographed at various time points. ImageJ software was used to measure gel area and assess collagen contraction.

## Migration assay

Met-1 (5 × 10$^4$) cells were placed into the upper chamber of 24 Transwell inserts, with pore sizes of 8 μm, in 300 μl NLF CM or ALF CM. Following 24 hr incubation, the upper side of the apical chamber was scraped gently with cotton swabs to remove nonmigrating cells, fixed with methanol, and stained with DAPI. Migrated cells were documented under a fluorescence microscope. ImageJ software was used to quantify migration.

## Multiplexed immunofluorescence staining

Fibroblast markers staining was performed in formalin-fixed paraffin-embedded (FFPE) blocks. Serial sections were obtained to ensure equal sampling of the examined specimens (5–10 μm trimming). FFPE sections from mouse lungs were deparaffinized and incubated in 10% neutral buffered formalin (NBF) for 20 min in room temperature, washed, and then antigen retrieval was performed with citrate buffer (pH 6.0; for αSMA and PDPN) or with Tris-EDTA buffer (pH 9.0; for S100A4). Slides were blocked with 10% BSA + 0.05% Tween20 and antibodies were used in a multiplexed manner with OPAL reagents, O.N. at 4°C (Opal Reagent pack and amplification diluent, Akoya Biosciences). Following overnight incubation with primary antibodies, slides were incubated with secondary antibodies conjugated to HRP for 10 min, washed, and incubated with OPAL reagents for 10 min. After each cycle, slides were stained sequentially with the next first antibody or finally with DAPI and mounted. Each antibody was validated and optimized separately, and the sequence of MxIF was optimized to confirm signals were not lost or changed during the multistep protocol. Slides were scanned at ×20 magnification using the Leica Aperio VERSA slide scanner. Quantitative analyses of fluorescence intensity were performed with ImageJ software.

For TF panel, lungs were fixed in PFA and embedded in O.C.T on dry ice. Serial sections were obtained to ensure equal sampling of the examined specimens (5 μm trimming). Sections were fixed with 4% PFA for 5 min, permeabilized by 0.2% Triton for 20 min, and fixed with NBF as described above. Antigen retrieval was performed using citrate buffer (pH 6.0). Slides were blocked with 1% BSA, 5% normal goat serum in 0.2% PBST for 1 hr, and primary antibody was incubated for O.N in 4°C. Slides were then incubated with secondary antibodies conjugated to HRP for 10 min and incubated with OPAL reagents for 10 min. We used the following staining sequences of primary antibodies: YFP, HSF1, STAT3, NFkB1, and MYC, and the fluorophores Opal 520, Opal 690, Opal 650, Opal 620, and Opal 570, respectively. The samples were imaged with a LeicaSP8 confocal laser-scanning microscope (Leica Microsystems, Mannheim, Germany).

## Flow cytometry analysis and cell sorting

Single-cell suspensions of lungs isolated from FVB/n;*Col1α1*-YFP or PyMT;*Col1α1*-YFP mice were stained using the following antibodies: anti-EpCAM-APC (eBioscience, 17-5791), anti-CD45-PerCP-Cy5.5 (eBioscience, 45-0451), and anti-CD31-PE-Cy7 (eBioscience, 25-0311). DAPI was used to exclude dead cells (Molecular Probes; D3571). Ki67-PE (BioLegend, 652403) intracellular staining of fibroblasts was done using an intracellular staining kit (BD Biosciences, 554714) according to the manufacturer's protocol. Flow cytometric analysis was performed using CytoFLEX Flow Cytometer (Beckman Coulter). Cell sorting was performed using BD FACSAria II or BD FACSAria Fusion (BD Biosciences). Data analysis was performed with the Kaluza Flow Analysis software (Beckman Coulter).

## RNA isolation and qRT-PCR

RNA from sorted cells was isolated using the EZ-RNAII kit (20-410-100, Biological Industries) according to the manufacturer's protocol. RNA from in vitro experiments was isolated using the PureLink RNA Mini Kit (Invitrogen; 12183018A). cDNA synthesis was conducted using qScript cDNA Synthesis kit (Quanta; 95047-100). Quantitative real-time PCRs (qRT-PCR) were conducted using PerfeCTa SYBR Green Fastmix ROX (Quanta; 95073-012). In all analyses, expression results were normalized to *Gusb or Gapdh* and to control cells. RQ (2$^{-\Delta\Delta Ct}$) was calculated.

## Transfection of primary fibroblasts

NLFs were cultured in DMEM supplemented with 10% FCS. At 70% confluency, cells were transfected with Accell Delivery Media (GE Dharmacon; B-005000) supplemented with 1 μM Accell SMARTpool mouse *Myc* siRNA (Dharmacon; E-040813) or Accell Control Pool nontargeting siRNA (Dharmacon; D-001910) for 96 hr. Accell SMARTpool contains a mixture of four siRNAs targeting one gene and provides extended duration of gene knockdown with only minimal effects on cell viability and the innate immune response. The efficiency of *Myc* siRNA knockdown was analyzed by qRT-PCR.

For individual siRNA experiments, NLFs were cultured and transfected as described, utilizing individual Myc targeting siRNA constructs (Dharmacon, A-040813-20, A-040813-18, A-040813-17) or control siRNA.

For overexpression of Myc, cells were transiently transfected with a plasmid of Myc (MGC Mouse Myc cDNA pCMV-SPORT6: mammalian expression insert sequence: BC006728, #MMM1013-202763479) or with a control plasmid. Cells were transfected with jetPRIME (polyplus transfection, 114-01) according to the manufacturer's protocol. All experiments were performed 24 hr following transfection.

XTT assay (Biological Industries, 20-300-1000) was performed 24 hr following transfection according to the manufacturer's protocol.

## RNA-seq

CD45⁻EpCAM⁻YFP⁺fibroblasts were isolated by cell sorting from normal FVB/n; *Col1a1*-YFP mice (n = 4), PyMT;*Col1a1*-YFP micrometastases-bearing mice (n = 3), and PyMT;*Col1a1*-YFP macrometastases-bearing mice (n = 4). Micrometastases were defined as visible mammary tumors, the absence of visible macrometastases, and the presence of EpCAM⁺ cells in lungs. Cells were collected into Trizol LS reagent (Life Technologies; 10296-028), and RNA was isolated according to the manufacturer's instructions. Transcriptomic sequencing of RNA was performed using NEBNext rRNA Depletion Kit (New England Biolabs, Inc; E6310S) and SMARTer Stranded Total RNA-Seq Kit – Pico Input (Clontech; 635005) and sequenced on the Illumina HiSeq 2500 sequencer (Illumina, Inc) at the Technion Genome Center. Sequenced reads were aligned to the mouse genome (mm9) using TopHat2 (*Kim et al., 2013*). Gene expression counts were calculated using HTseq-count (*Anders et al., 2015*) using Gencode annotations. Only genes that got at least 20 counts in at least three replicate samples were included in subsequent analysis (12,105 genes). Gene expression counts were normalized using quantile normalization (*Bolstad et al., 2003*). Levels below 20 were then set to 20 to reduce inflation of FC estimates for lowly expressed genes. Preliminary differential expression analysis was carried out using DESeq2 (*Love et al., 2014*). For subsequent analyses, only protein coding genes were included. In addition, coefficient of variance (CV) was calculated per group (NLF, MIF, MAF) and the top 1% most in-group deviated genes (top 1% CV) were excluded, leaving a total of 11,115 genes.

## Stage-specific signature analysis

The top altered genes from MAF vs. NLF were selected based on FC $\geq$ |2|. The MIF vs. NLF genes were selected based on an FC cutoff |1.5|. Data was Z-scored per gene. Venn diagram was generated using Bioinformatics and Evolutionary Genomics website (http://bioinformatics.psb.ugent.be/webtools/Venn/). All hierarchical clustering (based on Euclidean distance and average linkage) and PCAs were performed using JMP software version 14 and up.

### Gene selection based on network connectivity

Each group of genes (MIF vs. NLF, MAF vs. NLF, and MAF vs. MIF) were subjected to PPIs analysis using the STRING platform (*Szklarczyk et al., 2017*). The minimum confidence of interaction was defined as confidence $\geq$ 0.3 and connections based on text-mining were excluded. Groups of under four genes were excluded, narrowing the size of each group by ~50%.

## Pathway enrichment

For functional annotation, pathway and enrichment analysis, each comparison was analyzed separately, to a total of six comparisons (MIF vs. NLF up, MIF vs. NLF down, MAF vs. NLF up, MAF vs.

NLF down, MAF vs. MIF up, MAF vs. MIF down). Over-representation analysis was performed using the ConsensusPath DataBase (CPDB) (*Herwig et al., 2016*; *Kamburov et al., 2011*) (http://cpdb.molgen.mpg.de/MCPDB) platform for GO-molecular function (MF) and GO-biological process (BP), Reactome, and KEGG. Terms larger than 500 genes were excluded. Results were considered significant with a p-value<0.01, q-value < 0.05, and a coverage $\geq$ 3%. To increase the specificity of the enriched terms, we compared the relative overlap and the number of shared entities between the enriched terms from the three different databases (GO, KEGG, and Reactome). Selected terms with at least two shared entities and a relative overlap $\geq$ 0.2 were grouped and annotated based on a common enriched function. Groups smaller than three terms were excluded. These steps enabled the selection of the top ~10% most highly and significantly connected terms.

Bubble plot heat maps were generated with averaged log-transformed q-values [-Log$_{10}$(q-value)]. For terms enriched in a group of downregulated genes, the value of the average log-transformed q-value was transformed to a negative value by duplicating the average log-transformed value by (−1).

Heat maps were generated per annotation group, with a [log$_2$(Fold-change)] of gene expression calculated per comparison (MIF vs. NLF, MAF vs. NLF, and MAF vs. MIF).

## Gene Set Enrichment Analysis (GSEA)

The GSEA Java plug-in was used to probe log-transformed normalized expression data (*Subramanian et al., 2005*) (http://software.broadinstitute.org/gsea/index.jsp). Settings for the analysis were defined as follows: gene set database – hallmark gene sets only, number of permutations −1000, comparisons – each separately (MIF vs. NLF, MAF vs. NLF, MAF vs. MIF), permutation type – gene_set, minimum size – 5, maximum size – 500. Significant results were considered for false discovery rate (FDR) < 0.05 and normalized enrichment score (NES) > |2|.

## TF ranking

### Selection of TFs

TFs (*Hif1a, Hsf1, Myc, Nfkb1, Stat3*) that were enriched in pathway enrichment and/or GSEA analyses were selected as candidates and subjected to subsequent analyses.

### STRING

All five candidate TFs were subjected to PPI analysis in combination with each list of stage-specific genes per comparison (upregulated and downregulated in MIF vs. NLF, MAF vs. NLF, or MAF vs. MIF separately) using the STRING platform (*Szklarczyk et al., 2017*). The confidence of the interaction was defined as >0.2. For the ranking of each TF, the number of separate interactions for each TF was counted.

### Advanced Network Analysis Tool (ANAT)

The ANAT application (*Yosef et al., 2011*) was used as an add-in to Cytoscape (version 7 and up) software. We performed the analysis for each TF separately and for all of the TFs combined. The TFs were defined as anchors in the list, and the target genes were each list of stage-specific genes per comparison separately. An HTML report of all possible pathways between the anchor and each gene in the target genes list was generated. The minimum confidence for a connection was defined as confidence >0.2. An anchor could be connected to a target directly or indirectly. For the ranking of each TF, we calculated several parameters of the protein-protein network: (1) the number of stage-specific genes connected with each TF directly (1st neighbor), (2) the average shortest path for each TF, and (3) the centrality of the network. Parameters 2 and 3 were calculated using the network analysis tool of the Cytoscape software.

### RegNetwork

Each TF was defined as a regulator in the RegNetwork database (*Liu et al., 2015*). For ranking of each TF, the number of registered target genes from each list of stage-specific genes was counted.

### VarElect

VarElect platform (*Stelzer et al., 2016*) was utilized to analyze the relation of each list of stage-specific genes per comparison separately with each TF. Each gene from the list received a score according to its relation to the TF. For the ranking of each TF, several parameters were considered: (1) the number of directly related genes, (2) the average score of related genes, and (3) the average score of indirectly related genes.

### Ranking

Ranking parameters described above were Z-scored per parameter. For 'Characteristic path length,' results were first transformed with a ($-1$) power. Statistical analysis was performed using one-way ANOVA with Tukey correction for multiple comparisons.

## Human breast cancer data

The expression of the metastasis-associated gene signature and *MYC*, *NFKB1*, or *STAT3* was analyzed in human breast cancer stroma based on a publicly available dataset GSE14548 (*Ma et al., 2009*). Correlation analysis between *MYC* and its downstream genes derived from the metastasis-associated gene signature was performed on normalized expression values using Pearson correlation. p-value<0.05 was considered significant.

## Human MYC staining

Human patient samples were collected and processed at the Sheba Medical Center, Israel, under an approved Institutional Review Board (IRB) (3112-16). Sections stained for MYC were analyzed by an expert pathologist (Prof. Iris Barshack). Images were scanned at ×20 magnification using the Leica Aperio VERSA slide scanner. Analysis of the staining was performed using ImageScope software.

## Statistical analysis

Statistical analyses were performed using GraphPad Prism software and JMP pro 14 and 15 software. For two groups, statistical significance was calculated using t-test with Welch correction. For more than two comparisons, one-way ANOVA with Tukey correction for multiple comparisons was applied. All tests were two-tailed. p-value of $\leq 0.05$ was considered statistically significant. Correlation analyses were based on linear regression with Pearson correlation. Bar graphs represent mean and standard deviation (SD) unless otherwise stated. All experiments represent at least three separate biological repeats, unless otherwise stated.

## Data access

All raw and processed sequencing data generated in this study have been submitted to the NCBI Gene Expression Omnibus (GEO; http://www.ncbi.nlm.nih.gov/geo/) under accession number GSE128999.

## Acknowledgement

The authors thank Dr. Ran Elkon for his help with data analysis.

## Additional information

### Funding

| Funder | Grant reference number | Author |
| --- | --- | --- |
| H2020 European Research Council | 637069 MetCAF | Ophir Shani<br>Yael Raz |
| Israel Science Foundation | 1060/18 | Ophir Shani<br>Yael Raz<br>Noam Cohen<br>Neta Erez |
| The Emerson Collective | | Ophir Shani |

| | | Lea Monteran<br>Neta Erez |
|---|---|---|
| Israel Cancer Association | | Ophir Shani<br>Neta Erez |
| Israel Cancer Research Fund | Project Grant | Ophir Shani<br>Yael Raz<br>Lea Monteran<br>Neta Erez |
| Breast Cancer Research Foundation | | Or Megides<br>Ilan Tsarfaty |

The funders had no role in study design, data collection and interpretation, or the decision to submit the work for publication.

### Author contributions

Ophir Shani, Conceptualization, Data curation, Formal analysis, Investigation, Methodology, Writing - original draft, Project administration; Yael Raz, Conceptualization, Data curation, Formal analysis, Investigation, Methodology; Lea Monteran, Noam Cohen, Investigation; Ye'ela Scharff, Oshrat Levi-Galibov, Or Megides, Dana Silverbush, Formal analysis, Investigation; Hila Shacham, Formal analysis; Camilla Avivi, Resources, Investigation; Roded Sharan, Asaf Madi, Resources, Methodology; Ruth Scherz-Shouval, Resources; Iris Barshack, Resources, Formal analysis, Methodology; Ilan Tsarfaty, Conceptualization, Resources, Formal analysis, Supervision, Methodology, Writing - original draft, Project administration; Neta Erez, Conceptualization, Resources, Supervision, Investigation, Methodology, Writing - original draft, Project administration

### Author ORCIDs

Ophir Shani (iD) https://orcid.org/0000-0001-5002-6274
Neta Erez (iD) https://orcid.org/0000-0001-6506-9074

### Ethics

Human subjects: Human patient samples were collected and processed at the Sheba Medical Center, Israel under an approved institutional review board (IRB) (3112-16).
Animal experimentation: This study was performed in strict accordance with the recommendations in the Guide for the Care and Use of Laboratory Animals of the Tel Aviv University. All of the animals were handled according to approved institutional animal care and use committee (IACUC) protocols #: 01-18-035, M-13-026, 01-17-024 of the Tel Aviv University.

### Decision letter and Author response

Decision letter https://doi.org/10.7554/eLife.60745.sa1
Author response https://doi.org/10.7554/eLife.60745.sa2

## Additional files

### Supplementary files

• Supplementary file 1. Related to *Figure 3*. Detailed enrichment results for all comparisons based on selection criteria.

• Supplementary file 2. Related to *Figure 3*. Full Gene Set Enrichment Analysis results for all comparisons, false discovery rate < 0.05, normalized enrichment score > |2|.

• Supplementary file 3. Related to *Figure 4*. List of terms containing transcription factors enriched in all comparisons.

• Supplementary file 4. Related to *Figure 4*. Full results of transcription factor ranking of all comparisons.

• Transparent reporting form

## Data availability

Sequencing data have been deposited in GEO under accession code GSE128999.

The following dataset was generated:

| Author(s) | Year | Dataset title | Dataset URL | Database and Identifier |
|---|---|---|---|---|
| Shani O, Erez N | 2019 | RNA-seq profiling of fibroblasts isolated from two distinct lung metastases stages and from normal lungs | https://www.ncbi.nlm.nih.gov/geo/query/acc.cgi?acc=GSE128999 | NCBI Gene Expression Omnibus, GSE128999 |

The following previously published dataset was used:

| Author(s) | Year | Dataset title | Dataset URL | Database and Identifier |
|---|---|---|---|---|
| Ma X, Sgroi DC | 2009 | Gene Expression Profiling of Tumor Microenvironment during Breast Cancer Progression | https://www.ncbi.nlm.nih.gov/geo/query/acc.cgi?=GSE14548 | NCBI Gene Expression Omnibus, GSE14548 |

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
