## [Decision Letter]

**Acceptance summary:**

The authors elegantly demonstrated transcriptional dynamics and plasticity of fibroblasts in breast cancer metastasis formation, and position Myc as central player in this. With these findings, the contribution of the tumor microenvironment on the formation of a metastatic niche in breast cancer is better understood.

**Decision letter after peer review:**

Thank you for submitting your article "Evolution of fibroblasts in the lung metastatic microenvironment is driven by stage-specific transcriptional plasticity" for consideration by *eLife*. Your article has been reviewed by 3 peer reviewers, including Wilbert Zwart as the Reviewing Editor and Reviewer #1, and the evaluation has been overseen by Richard White as the Senior Editor. The following individual involved in review of your submission has agreed to reveal their identity: Mara Sherman (Reviewer #2).

The reviewers have discussed the reviews with one another and the Reviewing Editor has drafted this decision to help you prepare a revised submission.

In this manuscript by Shani and colleagues, the authors investigate fibroblast evolution during the progression of breast cancer metastasis to the lung. The transcriptional profiles of fibroblasts and CAFs in the lung metastatic niche will be an important resource for the field, as this analysis has not been performed in the context of spontaneous lung metastases from breast tumors in immune-competent hosts. The work is significant in providing insights into transcriptional alterations in fibroblasts during metastatic seeding, but also in defining a role for stromal MYC in establishment of the lung metastatic niche, which is novel and unexpected. The topic is of interest and approach is promising, but the work is under-developed in its current stage and needs revisions.

Essential revisions:

1. Key results from transcriptional profiling experiments should be validated by immunohistochemistry to show expression changes in fibroblasts within metastatic lesions. This is especially important for the MIF group given the confounding impact of normal fibroblasts-transcriptional profiling results may represent an average of normal fibroblasts and activated CAFs in the micrometastatic lesions, or may represent a field effect such that fibroblasts in the ostensibly normal lung tissue are partially activated. This is important to distinguish to better characterize alterations to the fibroblast compartment early in metastatic progression.

2. Immunohistochemistry for MYC should be repeated with a fibroblast co-stain in the human lung metastasis tissues.

3. The first four figures are essentially analysis of the RNAseq on the fibroblasts. However, independent validation of the RNAseq analysis in another cohort of mice is lacking. The authors should prepare additional RNA and analyze the levels of key genes identified the RNAseq by qRT-PCR. This is particularly critical for the micro-met fibroblasts as the PCA suggested no consistent change in transcriptome between normal and micro-met fibroblasts. Currently, the only validation that I could find for a change in micro-met fibroblasts is for Hsp90aa1 in Figure 5 and the change in Hspe1 shown in Figure 3c is not validated in Figure 5.

4. Related to the above point, what is really lacking is in situ staining for either exemplar proteins using antibodies or mRNA using RNAscope or similar in situ technology. This is important as there is heterogeneity in CAFs and the changes may only be occurring in a subset of fibroblasts within the metastases. I realize that the Col1a1-YFP strategy partially restricts the fibroblasts that they isolate, but there is still considerable scope for heterogeneity within this population of metastasis associated fibroblasts.

5. The plots in Figure 3b and Figure 3c do not seem consistent. For example, Figure 3b suggests that stress response genes are unchanged in macro-met fibroblasts, but they clearly seem to be changed in Figure 3c.

6. How do the authors determine what is a fibroblast in Figure 1a? And how do they select the regions for quantification? I don't disagree with the conclusions, but do they exclude the αSMA+ peri-vascular cells from the analysis of normal tissue? How do they account for the PDPN expression in alveolar type I cells in the normal lung? This shouldn't be compared to PDPN expression in fibroblasts.

7. Figure 6e should be removed as it is bulk analysis of tumor tissue and will be dominated by cancer cells.

8. It is great that the authors have access to some human lung met. tissue. The authors should perform multiplexed IHC to confirm that the myc staining is really in fibroblasts and not another elongated cell type. It would also help to clarify the issue of heterogeneity as many of the elongated cells don't express detectable myc.

9. How are the gel contraction and wound assays controlled for potential effects of myc depletion on fibroblast proliferation?

10. Multiple independent siRNA against myc should be employed.

11. Is myc gain of function sufficient to drive a fibroblast into a CAF-like state? If so, is it a more ECM remodeling, inflammatory, or stressed state?

12. The localization of Hsf1, Stat3, and NFkB in the lung with micro-mets and macro-mets would also worth looking at. Are these transcriptional factors localized in the same fibroblasts or different fibroblasts?

---

## [Author Response]

In this manuscript by Shani and colleagues, the authors investigate fibroblast evolution during the progression of breast cancer metastasis to the lung. The transcriptional profiles of fibroblasts and CAFs in the lung metastatic niche will be an important resource for the field, as this analysis has not been performed in the context of spontaneous lung metastases from breast tumors in immune-competent hosts. The work is significant in providing insights into transcriptional alterations in fibroblasts during metastatic seeding, but also in defining a role for stromal MYC in establishment of the lung metastatic niche, which is novel and unexpected. The topic is of interest and approach is promising, but the work is under-developed in its current stage and needs revisions.

We thank the reviewers for their insightful comments, and for acknowledging the novelty, importance and originality of our findings. We are grateful for the chance to improve our study and clarify the points that were raised. To address the reviewers’ critiques and suggestions, we performed additional experiments and modified our manuscript. As a result, the revised version of our manuscript contains new data and analyses that collectively strengthen the results and their implications.

Essential revisions:1. Key results from transcriptional profiling experiments should be validated by immunohistochemistry to show expression changes in fibroblasts within metastatic lesions. This is especially important for the MIF group given the confounding impact of normal fibroblasts-transcriptional profiling results may represent an average of normal fibroblasts and activated CAFs in the micrometastatic lesions, or may represent a field effect such that fibroblasts in the ostensibly normal lung tissue are partially activated. This is important to distinguish to better characterize alterations to the fibroblast compartment early in metastatic progression.

As suggested, we analyzed by immunohistochemistry two key factors that were upregulated specifically in the MIF group: Thbs1 (Thrombospondin-1) and HSP90, in combination with YFP staining for all fibroblasts. While both Thbs1 and HSP90AA1 were also expressed by other cells in addition to fibroblasts, staining validated the RNA-seq results and confirmed that both Thbs1 and HSP90AA1 are mainly upregulated in CAFs at the micrometastatic stage. Expectedly, not all YFP+ fibroblasts were Thbs1^+^ or HSP90AA1^+^, suggesting that MIFs are heterogeneous and contain multiple functional subpopulations. Representative images of staining were added to the revised version in Figure 3—figure supplement 2. Moreover, in response to a similar comment below (comments #3,4), we also validated the expression of selected candidates from the RNAseq by qRT-PCR of normal fibroblasts, MIF and MAF, as detailed in our response to comment #3 below. The results are shown in Figure 3—figure supplement 1 of the revised manuscript. We thank the reviewers for these suggestions.

2. Immunohistochemistry for MYC should be repeated with a fibroblast co-stain in the human lung metastasis tissues.

We appreciate that validating the MYC staining in human samples with a fibroblast co-stain would support our conclusions. However, due to the ongoing COVID-19 pandemic, our collaborators at the Sheba Medical Center were not able to perform additional staining for us in a timely manner. We would like to emphasize that our observation of MYC+ positive cells was assessed by an expert pathologist, Prof. Iris Barshack, who is also a co-author on our manuscript.

3. The first four figures are essentially analysis of the RNAseq on the fibroblasts. However, independent validation of the RNAseq analysis in another cohort of mice is lacking. The authors should prepare additional RNA and analyze the levels of key genes identified the RNAseq by qRT-PCR. This is particularly critical for the micro-met fibroblasts as the PCA suggested no consistent change in transcriptome between normal and micro-met fibroblasts. Currently, the only validation that I could find for a change in micro-met fibroblasts is for Hsp90aa1 in Figure 5 and the change in Hspe1 shown in Figure 3c is not validated in Figure 5.

To address this comment, we isolated fibroblasts from additional cohorts of mice (independent of the mice profiled by RNA-seq) to further confirm the results identified by the RNA-seq at different metastatic stages. We performed qRT-PCR to test the expression of key genes from identified pathways (stress response, ECM remodeling, and inflammation). Analysis confirmed the RNAseq results of genes that were found to be upregulated in micro- or macro-metastases associated fibroblasts. These results are included in Figure 3—figure supplement 2 in the revised manuscript. In addition, to address the concern regarding the discrepancy in the expression of *Hspe1* in micromet associated fibroblasts (MIFs), we repeated these validations with more mice (n=5/6 total per group) and confirmed that the expression of this gene is significantly upregulated in MIFs, in agreement with the RNA-seq results. We amended Figure 5B to reflect the new results, and stated the change in biological repeats in the figure legends. We thank the reviewers for this important comment.

4. Related to the above point, what is really lacking is in situ staining for either exemplar proteins using antibodies or mRNA using RNAscope or similar in situ technology. This is important as there is heterogeneity in CAFs and the changes may only be occurring in a subset of fibroblasts within the metastases. I realize that the Col1a1-YFP strategy partially restricts the fibroblasts that they isolate, but there is still considerable scope for heterogeneity within this population of metastasis associated fibroblasts.

See our response to comment #1 above.

5. The plots in Figure 3b and Figure 3c do not seem consistent. For example, Figure 3b suggests that stress response genes are unchanged in macro-met fibroblasts, but they clearly seem to be changed in Figure 3c.

The comparisons shown in Figure 3B and 3C are different: The analysis in Figure 3B compares *pathways* that were enriched and shows that the stress response pathway is significantly enriched in fibroblasts isolated from the micrometastases stage (MIF). Figure 3C shows the expression level of specific *genes* in the distinct metastatic stages compared with normal fibroblasts, or between the stages. While there is indeed a trend of upregulation in the expression of stressrelated genes in the macro stage compared to normal fibroblasts (Figure 3C, middle row), this increase was not significantly enriched in MAFs in the pathway analysis. Further confirming this, comparing the stress response gene signature in MIF vs. MAF fibroblasts (Figure 3C, bottom line), it is evident that this gene signature is downregulated in the macro stage as compared with the micro stage. We apologize that this was not sufficiently clear. We clarified this point in the revised manuscript (page 4).

6. How do the authors determine what is a fibroblast in Figure 1a? And how do they select the regions for quantification? I don't disagree with the conclusions, but do they exclude the αSMA+ peri-vascular cells from the analysis of normal tissue? How do they account for the PDPN expression in alveolar type I cells in the normal lung? This shouldn't be compared to PDPN expression in fibroblasts.

To exclude αSMA+ peri-vascular cells from the analysis we selected regions of interest for quantification that did not contain blood vessels. Regarding the quantification of PDPN: it is indeed known that PDPN is expressed also in alveolar type I cells. However, since excluding these areas was not technically feasible, we assumed that the expression in lung parenchymal cells was unchanged between the stages, and therefore the changes we may identify will be fibroblast related. Indeed, as clearly seen in the images and quantification, the changes in PDPN were not significant.

7. Figure 6e should be removed as it is bulk analysis of tumor tissue and will be dominated by cancer cells.

As requested, we have removed the KM plots from the revised manuscript.

8. It is great that the authors have access to some human lung met. tissue. The authors should perform multiplexed IHC to confirm that the myc staining is really in fibroblasts and not another elongated cell type. It would also help to clarify the issue of heterogeneity as many of the elongated cells don't express detectable myc.

We agree with the reviewers that multiplexed IHC in human lung metastasis would strengthen our observations and conclusions. However, due to the ongoing COVID-19 pandemic, our collaborators at the Sheba Medical Center were not able to perform additional staining for us in a timely manner. Moreover, performing MxIF requires extensive calibrations, consuming a large number of tissue sections, which is not possible with the rare tissue samples of human lung metastases that we were able to obtain. We would like to emphasize that our observations of MYC+ positive stromal cells was assessed by an expert pathologist, Prof. Iris Barshack, who is also a co-author on our manuscript. However, the reviewers are right in pointing out that it is possible that not all the fibroblastic cells in the human tissue sections that expressed Myc are fibroblasts, and therefore we revised our phrasing in the manuscript. The sentence describing the results now reads: “We found that MYC was expressed in a subset of lung metastasis-associated stromal cells (Figure 6E), suggesting that stromal upregulation of MYC plays a functional role in human lung metastasis”.

9. How are the gel contraction and wound assays controlled for potential effects of myc depletion on fibroblast proliferation?

We controlled for potential effects of Myc depletion on cell proliferation by performing both XTT assay and Ki67 staining. In both methods, the results indicated that the effect on CAF-like functions that we found following knockdown of Myc were not due to changes in their proliferation. These results were added to the revised manuscript (Figure 5—figure supplement 8,9) and clearly stated. We thank the reviewers for pointing this out.

10. Multiple independent siRNA against myc should be employed.

The siRNA that we used was not a single siRNA, but rather a mix of *different* siRNAs targeting sequences. The Accell SMARTpool that we used contains a mixture of four siRNAs targeting one gene and provides extended duration of gene knockdown with only minimal effects on cell viability and the innate immune response. This is stated in the methods section.

To address the concern regarding the specificity of the Myc targeting siRNA, and to rule out the risk of an off-target effect, we deconvoluted the SMARTpool of siRNA sequences and repeated the experiments with independent siRNAs to Myc, as requested.

Primary lung fibroblasts were transduced with three independent Myc-targeting siRNAs or with control siRNA. Analysis of the results confirmed knockdown of Myc expression in fibroblasts with all three independent siRNAs. Analysis of the CAF-like functions collagen contraction and wound closure revealed that all three independent siRNA significantly inhibited fibroblast wound closure and partially inhibited collagen contraction. These results were added as a supplement to Figure 5 in the revised manuscript.

We would like to emphasize that we also performed overexpression of Myc and showed that it was sufficient to enhance CAF-like functions, further supporting out observations from the knockdown experiments.

Therefore, we can now confirm that the results shown in the manuscript are specific to Myc.

Our findings reveal a novel central role for Myc in the rewiring of metastases-associated fibroblasts and emphasize the importance of understanding the early changes that underlie metastasis. Overall, the findings presented in this manuscript elucidate for the first time the dynamic co-evolution and the transcriptional rewiring of stromal cells during the multi-stage process of breast cancer metastasis.

11. Is myc gain of function sufficient to drive a fibroblast into a CAF-like state? If so, is it a more ECM remodeling, inflammatory, or stressed state?

To address this interesting question, we performed additional experiments and transduced normal lung fibroblasts to overexpress Myc in vitro. Analysis of CAF-like functions revealed that scratch wound closure, which was attenuated when Myc expression was knocked down, is significantly enhanced by overexpression of Myc, in a proliferation-independent manner. These results are now included in revised Figure 5I,J and Figure 5—figure supplement 10. However, collagen contraction was not enhanced by overexpression of Myc, although it was attenuated by its knockdown. Notably, Myc overexpression induced upregulation of its target genes BCAT1 and ODC1, that were also upregulated in the RNA-seq results. In addition, we found that multiple proinflammatory genes were upregulated by Myc overexpression (including CXCL1, CXCL2, CCL2 and IL-6). While these genes are not direct targets of Myc, they are known NF-κB target genes. Myc itself is a target of NF-κB (Grumont R, et al. Immunity 2004, PMID: 15345217, La Rosa FA et al., Mol Cell Biol. 1994, PMID: 8289784), and the two transcription factors also share target genes (Han H et al., Nucleic Acids Res. 2018, PMID: 29087512). These results are shown in revised Figure 5K. Thus, overexpression of Myc was sufficient to activate CAF-like functions including wound closure and expression of its target genes, as well as pro-inflammatory signaling in fibroblasts. We thank the reviewer for this excellent suggestion.

12. The localization of Hsf1, Stat3, and NFkB in the lung with micro-mets and macro-mets would also worth looking at. Are these transcriptional factors localized in the same fibroblasts or different fibroblasts?

To address this question, we performed multiplex immunofluorescent staining (MxIF) for YFP, and the transcription factors (TF) Myc, Stat3, NFkB1 and HSF1 in tissue sections of micro- and macrometastases. The staining revealed that while some of the fibroblasts co-expressed several TF (new Figure 4E, solid boxes), other CAF subpopulations expressed only Myc (dashed boxes). These results are shown in new Figure 4E.